

# Leveraging spatial textures, through machine learning, to identify aerosol and distinct cloud types from multispectral observations

Willem J. Marais[1], Robert E. Holz[1], Jeffrey S. Reid[2], and Rebecca M. Willett[3]

[1]Space Science Engineering Center, University of Wisconsin-Madison, Madison, Wisconsin, USA
[2]Marine Meteorology Division, Naval Research Laboratory, Monterey, California, USA
[3]Department of Statistics & Computer Science, University of Chicago, Illinois, USA

**Correspondence:** Willem J. Marais (willem.marais@ssec.wisc.edu)

**Abstract.** Current cloud and aerosol identification methods for multi-spectral radiometers, such as the Moderate Resolution Imaging Spectroradiometer (MODIS) and Visible Infrared Imaging Radiometer Suite (VIIRS), employ multi-channel spectral tests on individual pixels (i.e. field of views). The use of the spatial information in cloud and aerosol algorithms has been primarily statistical parameters such as non-uniformity tests of surrounding pixels with cloud classification provided by the

multi-spectral microphysical retrievals such as phase and cloud top height. With these methodologies there is uncertainty in identifying optically thick aerosols, since aerosols and clouds have similar spectral properties in coarse spectral-resolution measurements. Furthermore, identifying clouds regimes (e.g. stratiform, cumuliform) from just spectral measurements is difficult, since low-altitude cloud regimes have similar spectral properties. Recent advances in computer vision using deep neural networks provide a new opportunity to better leverage the coherent spatial information in multi-spectral imagery. Using a combi-

nation of machine learning techniques combined with a new methodology to create the necessary training data we demonstrate improvements in the discrimination between cloud and severe aerosols and an expanded capability to classify cloud types. The training labeled dataset was created from an adapted NASA Worldview platform that provides an efficient user interface to assemble a human labeled database of cloud and aerosol types. The Convolutional Neural Network (CNN) labeling accuracy of aerosols and cloud types was quantified using independent Cloud-Aerosol Lidar with Orthogonal Polarization (CALIOP)

and MODIS cloud and aerosol products. By harnessing CNNs with a unique labeled dataset, we demonstrate the improvement of the identification of aerosol and distinct cloud types from MODIS and VIIRS images compared to a per-pixel spectral and standard deviation thresholding method. The paper concludes with case studies that compare the CNN methodology results with the MODIS cloud and aerosol products.

## 1 Introduction & problem formulation

A benefit of polar-orbiting satellite passive radiometer instruments, such as the Moderate Resolution Imaging Spectroradiometer (MODIS), Visible Infrared Imaging Radiometer Suite (VIIRS) and the constellation of geostationary sensors, such as with the Advance Baseline Imager (ABI) and Advanced Himawari Imager (AHI), is their ability to resolve the spatial and spectral properties of clouds and aerosol features while providing global coverage. Consequently, the optical property measurements of aerosol particles and clouds from radiometer instruments, dominate both the atmospheric research and operational communi-





ties (Schueler et al., 2002; Salomonson et al., 1989; Klaes et al., 2013; Parkinson, 2013; Platnick et al., 2016; Levy et al., 2013;
Al-Saadi et al., 2005). E.g., cloud and aerosol measurements on a global scale are routinely employed in advance modeling
systems such as those contributing to the Cooperative for Aerosol Prediction multi-model ensemble (ICAP-MME) (Xian et al.,
2018) and the NASA Goddard Earth Observing System version 5 (GEOS-5) models (Molod et al., 2012).

Our work in this paper is motivated by two related atmospheric research needs:

1. Current data assimilation and climate applications depend on the accurate identification of optically thick aerosol from
radiometer images on global scales; Fig. 1 and 2 illustrate the difficulty in identifying optically thick aerosol and cloud
types which is due to the low spectral contrast between optically thick aerosol and water clouds in individual fields of
view (FOV).

        2. Climatological research can benefit from the identifications of major cloud regimes, as illustrated in Fig. 2, since me-
teorological conditions and major cloud regimes are related due to the strong correlations between cloud types and
atmospheric dynamics (Levy et al., 2013; Tselioudis et al., 2013; Evan et al., 2006; Jakob et al., 2005; Holz, 2002); very
little work has been done on cloud type identification from radiometer images.

What both these research needs have in common is that a practitioner uses contextual differences, such as spatial and spectral
properties (e.g. patterns and "color"), to identify optically thick aerosol and cloud types from imagery. E.g., from Fig. 1
and 2, a practitioner would use spatial texture to make distinction between optically thick aerosol, closed-stratiform and cirrus
clouds. Although practitioners can visually make these distinctions, current operational NASA cloud and aerosol products are
not able to reliably identify optically thick aerosol and cloud types. Specifically, illustrated by Fig. 1 and 2, 1) NASA cloud
products mistake optically thick aerosol for clouds, 2) the aerosol optical depth (AOD) retrieved by the NASA aerosol products
of optically thick aerosol are labeled as "bad" by the quality control flags and 3) from the NASA cloud optical properties
product it is unclear how to distinguish different cloud types other than making a distinction between water and ice clouds.
Consequently, in a climatological research project involving aerosols, most large impact optically thick aerosol will be excluded
in the research study if the project relies on the MODIS level-2 cloud and aerosol products; this problem stresses the importance
for the reliable identification of optically thick aerosol.

The underutilization of spatial information in radiometer images is major reason why NASA operational cloud and aerosol
products struggle to identify optically thick aerosol and restrain the identification of different cloud types; most NASA products
operate on primarily spectral information of individual pixels and some simple spatial analysis such as stand deviation of image
patches. This paper demonstrates that convolutional neural networks (CNNs) can increase the accuracy in identifying optically
thick aerosol and provide the ability to identify different cloud types.

Recent applications of statistical and machine learning (ML) methods in atmospheric remote sensing have demonstrated
improvements upon current methodologies in atmospheric science. E.g., neural networks have been employed to 1) approxi-
mate computationally demanding radiative transfer models to decrease computation time (Boukabara et al., 2019; Blackwell,
2005; Takenaka et al., 2011), 2) infer tropical cyclone intensity from microwave imagery (Wimmers et al., 2019), 3) infer
cloud vertical structures and cirrus/high-altitude cloud optical depths from MODIS imagery (Leinonen et al., 2019; Minnis





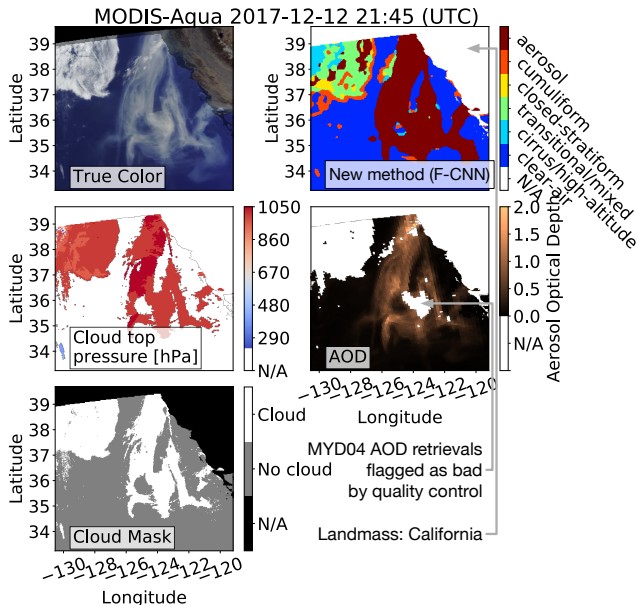

**Figure 1.** These images show an example where the MODIS spectral and standard deviation thresholding techniques (SSDT) misclassifies a thick smoke plume as a cloud; the smoke plume originated from Oregon and measurements were taken on 12th of Dec 2017 with Aqua-MODIS. According to the MODIS cloud top pressure (MYD06) and mask (MYD35) products, the whole smoke plume is labeled as a cloud. The MYD04 aerosol product associates the optically thick part of the smoke plume with "bad" quality control flags, since the SSDT technique employed by the MYD04 product is tuned to aggressively ignore any observations that has a close resemblance of a cloud. The new methodology uses a convolutional neural network to extract spatial texture features which enables the method to make an accurate distinction between aerosols that have smooth surfaces and clouds that have non-smooth surfaces.

et al., 2016) and 4) predict the formation of large hailstones from land-based radar imagery (Gagne II et al., 2019). Specific to

cloud and volcanic ash detection from radiometer images, Bayesian inference has been employed where the posterior distribution functions were empirically generated using hand labeled (Pavolonis et al., 2015) or coincident Cloud-Aerosol Lidar with Orthogonal Polarization (CALIOP) observations (Heidinger et al., 2016, 2012) or from a scientific product (Merchant et al., 2005).

## 1.1 Objectives

Inspired by the recent successful applications of ML in atmospheric remote sensing, we hypothesize that a methodology can be developed that accurately identify optically thick aerosol and cloud types by better utilizing the coherent spatial information in moderate-resolution multi-spectral imagery via recent advances in CNN architectures. Leveraging these recent advances we explore the following questions:



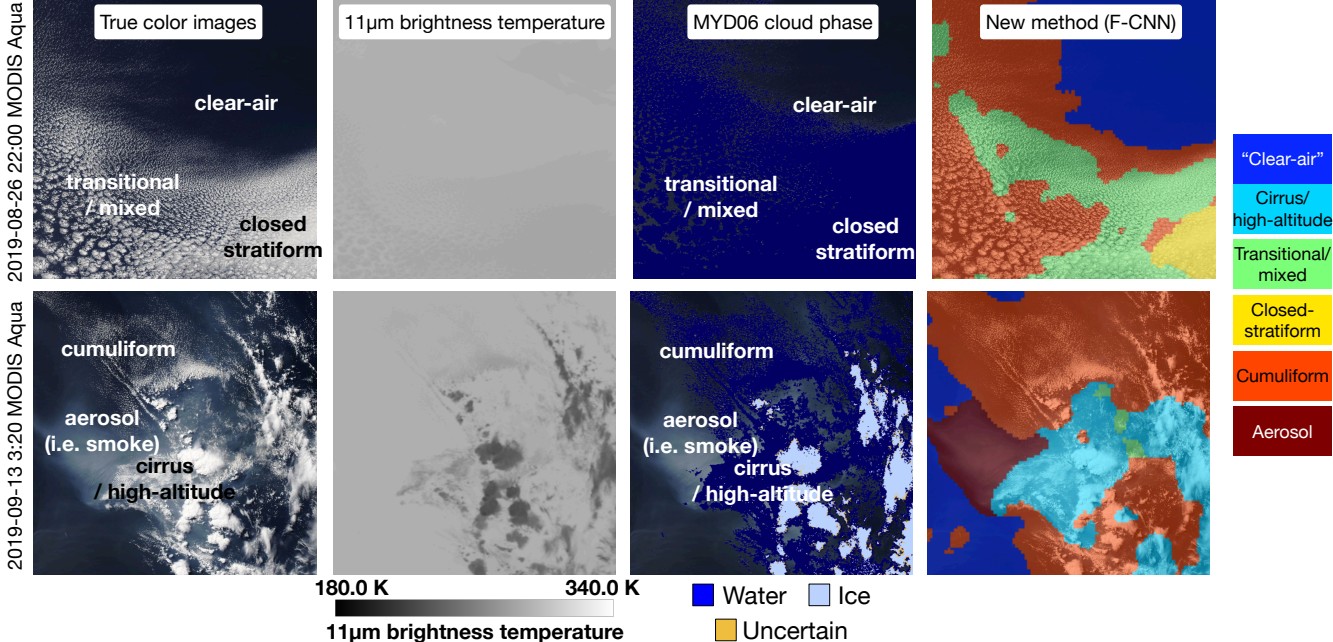

**Figure 2.** These images show the different texture smoothness properties of the stratiform, cumuliform and cirrus/high-altitude cloud at different spectral channels and how the proposed methodology (i.e. the new method) was able to leverage the smoothness properties to make a distinction between different cloud types. Stratiform clouds are smoother than cumuliform clouds, and both stratiform and cumuliform clouds are smoother than cirrus/high-altitude clouds at $11\mu m$ brightness temperature. From MYD06 cloud phase product it is unclear how to set different cloud types apart other than cirrus/high altitude clouds.

1. Can the application of CNNs with training data developed from a human classified dataset improve the distinction between cloud and optically thick aerosol events and provide additional characterization of clouds leveraging the contextual information to retrieve cloud type?

2. When identifying aerosol and cloud types by dividing up an image into smaller patches and extracting spatial information from the patches, what is the optimal patch size (number of pixels) that provides the best identification accuracy of the aerosols and cloud types?

3. What is the performance of the aerosol and cloud type identification when compared to lidar observations such as CALIOP?

We specifically consider a human labeled dataset instead of using an active instrument such as CALIOP and Cloud Profiling Radar (CPR) to produce a labeled dataset, since the cloud and aerosol products of these instruments have limitations; e.g., CPR only accurate detect precipitating clouds, and both CALIOP and CPR are nadir viewing instruments with which only a fraction MODIS/VIIRS pixels can be labelled (Kim et al., 2013; Stephens et al., 2002).





## 1.2 Proposed methodology

To address the proposed questions above we developed a methodology in which we adopted a CNN to exploit both the spatial and spectral information provided by MODIS/VIIRS observations to make a distinction between basic categories of aerosol and cloud fields: "clear-air", optically thick aerosol features, cumuliform, transitional, closed-stratiform and cirrus/high-altitude

clouds. The capability to globally detect and monitor these cloud and aerosol types provides important new information about the atmosphere since these cloud types are strongly correlated with atmospheric dynamics (Tselioudis et al., 2013). This paper does not address identifying clouds by their specific forcing or physics; this is a topic of a subsequent paper. Here, we are only interested in differentiating spatial cloud patterns and major aerosol events.

To conceptually demonstrate the new capabilities, Fig. 1 and 2 show the results of the methodology which has identified 1)

an optically thick aloft smoke plume and 2) cloud classification based on the contextual information. Notice the MODIS cloud top pressure (MYD06) and mask products (MYD35) misclassified the smoke plume as a cloud and the MODIS AOD retrievals are labeled as "bad" by the MYD04 quality control (Frey et al., 2008; Levy et al., 2013)[1].

The advantage of the proposed methodology is that it provides a better approach to leveraging the contextual information in the imagery to separate clouds from optically thick aerosol features using the differences in spatial structure. Although

the MODIS cloud and aerosol algorithms, which are based on per-pixel spectral and standard deviation (SD) thresholding techniques, correctly identifies clouds overall, simple spatial statistics on different spectral images do not provide enough independent information to uniquely separate optically thick aerosol events from cloud (Platnick et al., 2016; Levy et al., 2013).

A focus of the proposed methodology is the creation of a training dataset that enables a CNN to extract spatial features of

aerosols and cloud types. We adapted the NASA Worldview web-framework that enables an atmospheric scientist to hand-label aerosols and cloud types (NASA, 2020); Fig. 4 shows an example of the adapted user interface.

The work that we present in this paper has been primarily developed for daytime deep-ocean observations, since the identification of aerosols and cloud types over land and during the nighttime is a separate and challenging research project.

## 1.3 Outline

The outline of the paper is as follows. In Sect. 2 we described the proposed methodology to identify aerosols and cloud-types via a CNN. Included in Sect. 2 we 1) discuss why CNNs are capable of extracting spatial texture features from images, 2) explain why we use a pre-trained CNN rather than training a CNN from radiometer images and 3) explain why the pre-trained CNN extracts useful spatial texture features from radiometer images. In Sect. 3 we share test results of the proposed methodology using case studies. The paper is concluded in Sect. 4 with a discussion of future work.

---

[1]The smoke plume originated from Oregon and measurements were taken on 12th of Dec 2017 with Aqua-MODIS.



### 1.4 Acronyms and notation

Table 1 shows commonly used acronyms that are being used throughout this paper which the reader can reference for the sake of convenience.

| Acronym | Elaboration |
| --- | --- |
| AOD | Aerosol Optical Depth |
| CNN | Convolutional Neural Network |
| P-CNN | Pre-trained CNN |
| F-CNN | Fine-tuned CNN |
| FE | Feature Extractor |
| OD | Optical Depth |

**Table 1.** Commonly used acronyms that are being used throughout this paper.

## 2 Methodology

In this paper we define a method as a feature extractor (FE) algorithm with a multi-class classifier; the FE-algorithm extracts spatial and spectral information from the input images that enables the distinction between different image types by the multi-class classifier. The CNN, which our methodology employs, simultaneously extracts spatial information from the $0.642\mu$m and $0.555\mu$m solar reflectances and the $11\mu$m brightness temperature (BT) measurements where each of the bands jointly provide key information to identify aerosols and distinct cloud types.

For a given method (FE and classifier), cloud types and aerosols in MODIS/VIIRS image are identified by 1) dividing the input image into patches and 2) the method is applied to each patch to produce a label of "clear-air", optically thick aerosol, low-mid level cumuliform, transitional/mixed (transition between the two or mixed types), closed-stratiform or cirrus/high-altitude cloud. The size of the patches, from which the spatial features are extracted, determines the labeling image resolution; we show in the results how labeling accuracy is a function of the patch size. In this paper we consider patch sizes of 25 and 100 pixels where 1 pixel equals approximately 1 km.

Since it is not clear which FE-algorithm will yield spatial textual features that optimize the classification performance for identifying cloud types and aerosol, we consider three different FE-algorithms: 1) pre-trained CNN, 2) fine-tuned CNN and 3) traditional mean-standard deviation (MeanStd) FE-algorithm as a control. We employ a pre-trained CNN since a CNN requires a large training dataset (e.g. million images) to estimate its parameters (Simonyan and Zisserman, 2014), where the pre-trained CNN has its parameters pre-estimated from a different image classification problem (e.g. general internet images). A fine-tuned CNN has its parameters estimated from radiometer images where the initial values of the CNN parameters are from a pre-trained CNN. The main purpose of the MeanStd FE is to set a baseline result.





The next subsection introduces the algorithms that estimate the parameters of the methods and apply the methods on radiometer images. In Sect. 2.2 we discuss how the labeled dataset is created, which is sub-divided into a training and test dataset; from the test dataset a test (generalization) error is computed with which rank the different methods (Friedman et al., 2001). Sect. 2.3 gives a short overview of CNNs and elaborates on the pre-trained and fine-tuned CNNs. Sect. 2.5 provide details of the pre-trained CNN, fine-tuned CNN and MeanStd FE-algorithms.

## 2.1 Algorithms that train and apply methods

Two algorithms are used to 1) estimate the multi-class classifier parameters (i.e. training phase) of each method (i.e. multi-class classifier with a FE-algorithm) and 2) apply each method to a MODIS/VIIRS image to label the pixels in the image as aerosol and cloud type. The first algorithm estimates a method's parameters from the labeled training datasets of patches, and the test error is computed from the test labeled datasets of patches; Fig. 3A gives a pictorial overview and Alg. 1 gives a detailed outline of the first algorithm. The second algorithm applies a method (trained via Alg. 1) on a MODIS/VIIRS image by dividing the input image into patches and applying the method on each patch; Fig. 3B gives a pictorial overview and Alg. 2 gives a detailed outline of the second algorithm.

### 2.1.1 Algorithm 1 - Estimate method parameters - training phase

---

**Algorithm 1** Estimate method parameters - training phase

---

**Input:** (i) Training dataset consists of $K$ patches and labels denoted by $\{X^{(k)}\}_{k=1}^{K}$ and $\{Y^{(k)}\}_{k=1}^{K}$, respectively.

**Input:** (ii) Test dataset consists of $L$ patches and labels denoted by $\{\tilde{X}^{(l)}\}_{l=1}^{L}$ and $\{\tilde{Y}^{(l)}\}_{l=1}^{L}$, respectively.

**Input:** (iii) A `feature_extractor` algorithm such as FE-Alg. I, II or III (refer to Sect. 2.5).

**Assume:** The size (length and width) of each patch is either 100 or 25 pixels with wavelengths 0.642 $\mu$m, 0.555 $\mu$m and 11 $\mu$m.

  1: **for** $k = 1, 2, \ldots, K$ **do**

  2:      /* Extract feature with feature extractor algorithm */

  3:      $Z^{(k)} = $ `feature_extractor` $(X^{(k)})$

  4: **end for**

  5: $\hat{W} = $ Use Alg. 1a to estimate parameters of multinomial classifier with training features vectors $\{Z^{(k)}\}_{k=1}^{K}$ and labels $\{Y^{(k)}\}_{k=1}^{K}$.

  6: **for** $l = 1, 2, \ldots, L$ **do**

  7:      /* Extract feature with feature extractor algorithm */

  8:      $\tilde{Z}^{(l)} = $ `feature_extractor` $(\tilde{X}^{(l)})$

  9: **end for**

10: Compute test error with (5): Err $\left( \{\tilde{Z}^{(l)}\}, \{\tilde{Y}^{(l)}\}, \hat{W} \right)$

11: **return** `classifier` $(\hat{W}) = $ trained multinomial classifier

---

We give a summary of how Alg. 1 trains the multi-class classifier, and in the rest of the subsection we provide the necessary details of the algorithm. Alg. 1 takes as input the labeled training and test datasets of patches with the given FE-algorithm and





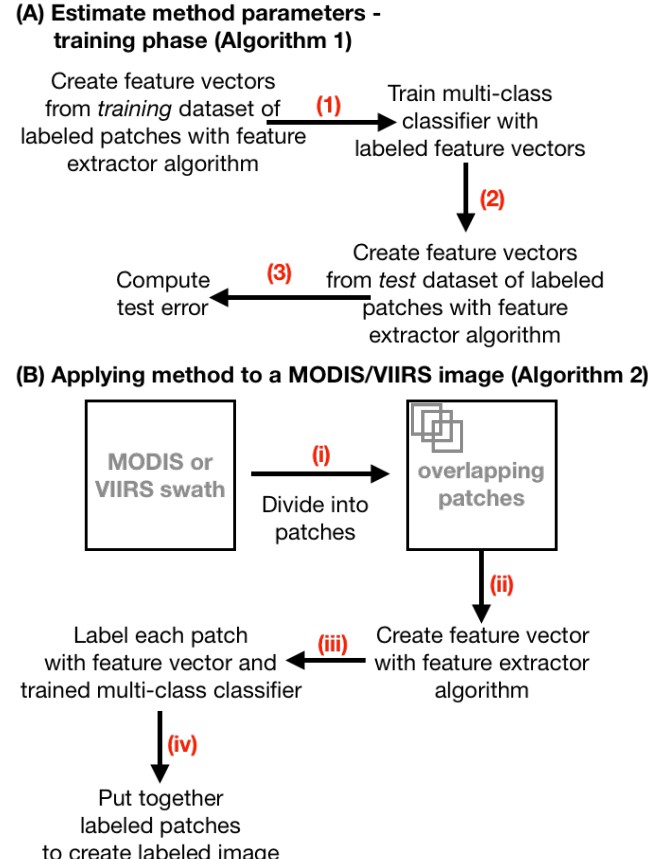

**Figure 3.** A method that identifies cloud types and aerosols in MODIS/VIIRS images consists of a feature extractor (FE)-algorithm and a multi-class classifier. (A) The parameters of the method's multi-class classifier are estimated and the test error is computed by 1) creating the feature vectors from each patch in the training dataset via the method's FE-algorithm, 2) the multi-class classifier's parameters are estimated using the training feature vectors and the corresponding labels, 3) the feature vectors from each patch in the test dataset are extracted and the test error is computed from the test feature vectors and corresponding labels. (B) A trained method is applied on a MODIS/VIIRS image by i) dividing the MODIS/VIIRS image into overlapping patches, ii) for each patch the spatial texture feature vectors are extracted via the method's FE-algorithm, iii) each patch is labeled via the feature vector and the method's trained multi-class classifier and iv) all the labeled patches are put together to create a labeled image.

the output is a trained multinomial classifier (i.e. multi-class classifier) (Krishnapuram et al., 2005; Friedman et al., 2001). For each patch the multinomial classifier models a probability value for each label. This probability model has several parameters

that are estimated from the labeled training dataset by finding the parameters that minimizes an objective (i.e. cost) function; the objective function consists of 1) a loss function that "fits" the parameters onto the patches and labels, and 2) a $l_2$ (euclidean norm) penalty function with a tuning parameter that regularizes the parameters to prevent the multinomial classifier from





"overfitting" on the training dataset (Friedman et al., 2001). The optimum tuning parameter is estimated via five-fold cross-validation, since the size of our training dataset is limited (Friedman et al., 2001).

The labeled training and test datasets are denoted by $\{X^{(k)}\}_{k=1}^{K} \times \{Y^{(k)}\}_{k=1}^{K}$ and $\{\tilde{X}^{(l)}\}_{l=1}^{L} \times \{\tilde{Y}^{(l)}\}_{l=1}^{L}$, respectively, where the superscripts $(k)$ and $(l)$ are indices to the different patches and labels. Each patch, denoted by 3-dimensional tensor $X \in \mathbb{R}^{S \times S \times 3}$, have three layers and covers an area of $S$ by $S$ pixels. The corresponding label vector $Y \in \mathbb{R}^6$ is a six element canonical vector (i.e. $i$th vector equals one and the rest are zero) where the position of the one-element corresponds to either category "clear-air", optically thick aerosol features, cumuliform, transitional, closed-stratiform or cirrus/high-altitude clouds.

For each patch in the training dataset the FE-algorithm creates $\tilde{S}$ element feature vectors denoted by $Z \in \mathbb{R}^{\tilde{S}}$. The probability value of a feature vector $Z$ being of label $Y$ is modeled by normalized exponential function

$$P(W; Z, Y) = \frac{\exp\left(Z^T W Y\right)}{\sum_{i=1}^{6} \exp\left(Z^T W e_i\right)}, \tag{1}$$

where $W \in \mathbb{R}^{\tilde{S} \times 6}$ is the parameter of the multinomial classifier which linearly weights the feature vector $Z$, the label vector $Y$ selects the column of $W$ that is multiplied with the feature vector $Z$ and $e_i \in \mathbb{R}^6$ is a six element canonical vector.

Alg. 1 employs Alg. 1a to estimate the optimum weight matrix $W$. In more detail; with the training features and label vectors $\mathcal{Z} = \{Z^{(k)}\}_{k=1}^{K}$ and $\mathcal{Y} = \{Y^{(k)}\}_{k=1}^{K}$, Alg. 1a estimates the weight matrix $W$ by minimizing objective function

$$\Phi(W; \mathcal{Z}, \mathcal{Y}, \lambda) = \overbrace{\ell(W; \mathcal{Z}, \mathcal{Y})}^{\text{loss function}} + \underbrace{\lambda}_{\text{tuning parameter}} \times \overbrace{\text{trace}\left(W^T W\right)}^{\text{penalty term}}, \tag{2}$$

$$\ell(W; \mathcal{Z}, \mathcal{Y}) = \sum_{(Z, Y) \in \mathcal{Z} \times \mathcal{Y}} -\log_e P(W; Z, Y), \tag{3}$$

where (3) is the negative log-likelihood (i.e. loss) function and $\lambda > 0$ is the regularizer (i.e. tuning) parameter of the $l_2$ penalty term as indicated in (2). The optimum tuning parameter $\lambda$ is estimated by Alg. 1a using five-fold cross-validation (Friedman et al., 2001), where for each tuning parameter a validation error is computed and the optimum tuning parameter corresponds to the smallest validation error; the validation error is computed using the loss function (3).

A patch is labeled with the function

$$\Psi(Z; \hat{W}) = e_{i^*} \text{ where } i^* = \arg\max_{i \in \{1, 2, \dots, 6\}} P(\hat{W}; Z, e_i), \tag{4}$$

which produces a label vector by finding the canonical vector $e_i$ that maximizes the modeled probability (1) with the extracted feature vector $Z$ and estimated weight matrix $\hat{W}$.

For a given test datasets of feature $\tilde{\mathcal{Z}} = \{\tilde{Z}^{(l)}\}_{l=1}^{L}$ and label $\tilde{\mathcal{Y}} = \{\tilde{Y}^{(l)}\}_{l=1}^{L}$ vectors the test error is computed by

$$\text{Err}\left(\tilde{\mathcal{Z}}, \tilde{\mathcal{Y}}, \hat{W}\right) = \frac{1}{\left|\tilde{\mathcal{Y}}\right|} \sum_{(\tilde{Z}, \tilde{Y}) \in \tilde{\mathcal{Z}} \times \tilde{\mathcal{Y}}} \mathbb{1}\left\{\tilde{Y} \neq \Psi(\tilde{Z}; \hat{W})\right\}, \tag{5}$$





where $\left|\tilde{\mathcal{Y}}\right|$ is number of label vectors and $\mathbb{1}\{\cdot\}$ is an indicator function defined as

$$\mathbb{1}\left\{\tilde{Y} \neq \Psi(\tilde{Z}; \hat{W})\right\} = \begin{cases} 1 & \text{if } \tilde{Y} \neq \Psi(\tilde{Z}; \hat{W}) \\ 0 & \text{otherwise.} \end{cases} \tag{6}$$

Alg. 1 assumes that the parameters of the CNN FE-algorithm are already estimated. Hence, the parameters of the fine-tuned CNN are estimated separately from Alg. 1 with the same training dataset that is used by Alg. 1; the fine-tuning is accomplished using a software package such as Tensorflow (Abadi et al., 2016). The software package `scikit-learn` can be used to minimizes the objective function (2) (Pedregosa et al., 2011).

---

**Algorithm 1a** Estimate parameters of multinomial classifier (i.e. multi-class classifier) via 5-fold cross-validation

---

**Input:** Features vectors $\mathcal{Z} = \{Z_k\}_{k=1}^{K}$ with labels $\mathcal{Y} = \{Y_k\}_{k=1}^{K}$.

1: Shuffle the order of the feature vectors $\mathcal{Z}$ and corresponding labels $\mathcal{Y}$.

2: Partition the sets $\mathcal{Z}$ and $\mathcal{Y}$ into five equal partitions which are denoted by $\mathcal{P}_r[\mathcal{Z}]$ and $\mathcal{P}_r[\mathcal{Y}]$, where $r \in \{1, 2, 3, 4, 5\}$.

3: Set the multinomial classifier tuning parameters $\{\lambda_d\}_{d=1}^{D} = \{10^{-2}, \ldots, 10^{2}\}$ where $\log_{10} \lambda_d = -2 + 4(d-1)/(D-1)$.

4: For $d = 1, 2, \ldots, D$ define $e_d = 0$ as the negative log-likelihood validation value which will be produced by the multinomial classifier loss function.

5: **for** $r = 1, 2, 3, 4, 5$ **do**

6:     /* Hold out the partition $\mathcal{P}_r$ to create the training dataset of the cross-validation $r$-th iteration */

7:     Define $\overline{\mathcal{P}}_r[\mathcal{Z}] = \cup_{q \neq r} \mathcal{P}_q[\mathcal{Z}]$ and $\overline{\mathcal{P}}_r[\mathcal{Y}] = \cup_{q \neq r} \mathcal{P}_q[\mathcal{Y}]$.

8:     /* For each tuning parameter $\lambda_d$ compute the negative log-likelihood validation value */

9:     **for** $d = 1, 2, \ldots, D$ **do**

10:         /* Estimate parameters of multinomial classifier by minimizing (2) */

$$\hat{W} = \arg\min_{W} \Phi\left(W; \overline{\mathcal{P}}_r[\mathcal{Z}], \overline{\mathcal{P}}_r[\mathcal{Y}], \lambda_d\right)$$

11:         /* Compute validation error using the multinomial negative log-likelihood function (3) */

$$e_d = e_d + \ell\left(\hat{W}; \mathcal{P}_r[\mathcal{Z}], \mathcal{P}_r[\mathcal{Y}]\right)$$

12:     **end for**

13: **end for**

14: Choose optimum tuning parameter $\lambda^* = \lambda_{d^*}$ where $d^* = \arg\min_d e_d$.

15: /* Infer the optimum multinomial classifier parameter */

$$\hat{W} = \arg\min_{W} \Phi\left(W; \mathcal{Z}, \mathcal{Y}, \lambda^*\right)$$

16: **return** The trained multi-class classifier with its parameter vector $\hat{W}$.

---





### 185  2.1.2   Algorithm 2 - Applying trained method to label a MODIS/VIIRS image

Alg. 2 takes as input a MODIS/VIIRS image, patch size and a trained method which consists of a FE-algorithm with its corresponding trained multinomial classifier. For every pixel in the input MODIS/VIIRS image, Alg. 2 assigns a label to the

---

**Algorithm 2** Applying method to a MODIS/VIIRS image

---

**Input:** (i) Patch size $S$ (e.g. 100 or 25 pixels).

**Input:** (ii) The image $\mathbf{X} \in \mathbb{R}^{N \times K \times 3}$ that covers an area of $N$ by $K$ pixels with wavelengths 0.642 $\mu$m, 0.555 $\mu$m and 11 $\mu$m.

**Input:** (iii) The `feature_extractor` algorithm where the possible FE-algorithms are FE-Alg. I, II or III (refer to §2.5).

**Input:** (iv) The multinomial classifier, with its weight matrix $W$, which was trained by Algorithm 1 with the specific `feature_extractor`.

1:  $\mathcal{X}$ = Divide $\mathbf{X}$ into overlapping $S$ by $S$ pixel patches, where the stride between the patches is five pixels.

2:  Create a probability image $U \in \mathbb{R}^{N \times K \times 6}$ that covers an area of $N$ by $K$ pixels and has six layers.

3:  /* Iterate over rows and columns of input image $\mathbf{X}$ */

4:  **for** $n = 1, 2, \ldots, N$ and $k = 1, 2, \ldots, K$ **do**

5:     $\tilde{\mathcal{X}}$ = find the patches in $\mathcal{X}$ that overlaps the pixel and place the overlapping patches in the set $\tilde{\mathcal{X}} \subset \mathcal{X}$.

6:     $V = |\tilde{\mathcal{X}}|$ represent the number of overlapping patches.

7:     **for** every $X$ in $\tilde{\mathcal{X}}$ **do**

8:        /* Create feature vector */

9:        $Z =$ `feature_extractor`$(X)$

10:       /* Add probability value of each label to probability image $U$ using probability model (1) */

11:       **for** $i = 1, 2, \ldots, 6$ **do**

12:          /* Weight probability value with number of overlapping patches */

13:          $U_{n,k,i} = U_{n,k,i} + P(\hat{W}; Z, e_i)/V$

14:       **end for**

15:     **end for**

16: **end for**

17: Create label image $Y \in \mathbb{R}^{N \times K}$ that covers an area of $N$ by $K$ pixels.

18: **for** $n = 1, 2, \ldots, N$ and $k = 1, 2, \ldots, K$ **do**

19:     $Y_{n,k} = i^*$ where $i^* = \arg\max_{i \in \{1,2,\ldots,6\}} U_{n,k,i}$

20: **end for**

21: **return** Labeled image $Y$

---

pixel by 1) computing what is the probability that pixel belongs to one of the six label categories (e.g. "clear-air", optically thick aerosol, etc.) and 2) the label with maximum probability value is assigned to the corresponding pixel. To assign a probability value of a specific label to a pixel, Alg. 2 in summary 1) takes all the patches that intersects with the target pixel, 2) extract the feature vectors from the patches with the given FE-algorithm, 3) apply the multinomial classifier on each feature vector to produce a series of probability values (of the specific label) and 4) average the series of probability values to produce a final





probability value of the target pixel. This labeling procedure assumes that the category to which a pixel belongs is correlated with the categories of the surrounding pixels, implying that all patches that intersects a pixel contributes to the probability value
of the pixel.

In order to reduce the computational demand on estimating the probability values per pixel, the overlapping patches have a row and column index stride of five.

## 2.2   The labeled dataset

The labeled dataset, from which each method's parameters and error performance are estimated, is created through a Space
Science Engineering Center (SSEC) adapted NASA Worldview web-interface; the website allows an atmospheric scientist to interactively create a labeled dataset. Fig. 4 shows how the labeled dataset is assembled; the subset MODIS/VIIRS images that are 1) chosen by the atmospheric scientist 2) are divided into overlapping patches of the two different sizes of 25 and 100 pixels. From the labeled dataset a training and a test dataset are created by sampling from labeled datasets; the training and test datasets are used to estimate the parameters and error performance of a method, respectively (Friedman et al., 2001).

In order to ensure consistent labels for the study, one co-author created the labeled dataset (JSR); the system is able to track individual trainers to identify areas of ambiguity. The labeled dataset provides specific cloud and aerosol categories. The cloud categories are 1) "clear-air", 2) cirrus/high-altitude, 3) transitional/mixed, 4) closed-stratiform and 5) cumuliform, and the aerosol categories are 1) severe dust, 2) severe smoke and 3) severe pollution. For this study all the aerosol categories are aggregated into one aerosol label; our methodology does not identify specific aerosol types, because the required number of
samples in the training dataset increases exponentially as function of the number of distinct labels the multi-class classifier with the CNN FE-algorithm have to identify (Shalev-Shwartz and Ben-David, 2014).

The labeled subset of MODIS/VIIRS images in the datasets were projected onto an equilrectangular grid (i.e. equidistant cylindrical map projection) with a spatial resolution of 1 km (i.e. 1 pixel equals 1 km), and each solar reflectance channel was corrected for Rayleigh scattering (Vermote and Vermeulen, 1999).

### 2.2.1   Training and test datasets

For a specific patch size we created a training dataset by randomly sampling, without replacement, patches from the labeled dataset. The training dataset consists of 75% of the extracted patches, and the rest of the patches that were not sampled are placed into the test dataset. To prevent a multi-class classifier with a FE-algorithm to be biased towards classifying a specific cloud type or aerosol the random sampling of the patches were done such that 1) there is an equal portion of patches of each
label and 2) there is an equal portion of aerosol sub-types (e.g. smoke, dust, pollution) (Japkowicz and Stephen, 2002).

### 2.2.2   Quality assurance of labeled data

To label a subset of a MODIS/VIIRS image through the web-interface, as shown in Fig. 4, the atmospheric scientist 1) first selects the cloud and aerosol categories and 2) selects the region of the cloud and aerosol type by dragging a rectangular





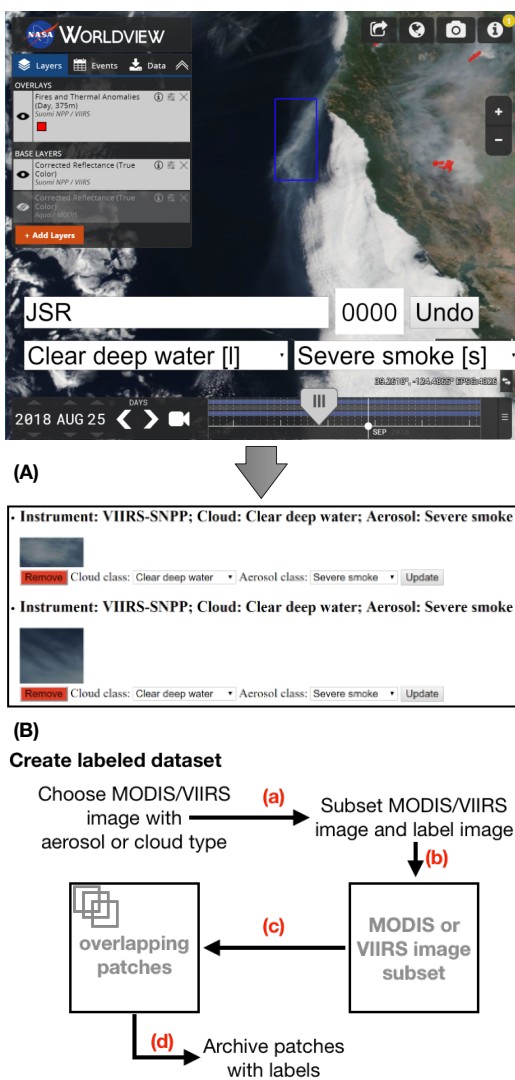

**Figure 4.** (A) The web-interface through which an atmospheric scientist can create a labeled dataset of aerosols and cloud types. First the user decides what to label; e.g. in the above image the user chose to label cumuliform clouds with a dust event. Then the user selects the region of interest (the blue rectangle over the aerosols event) via the mouse cursor; a counter increases (0002) to indicate the labeled image has been archived in a database. The user has the option of "undo" the most recent label image, and the user can review the labeled images by either removing or re-labeling the images. (B) To create the labeled dataset (a) the atmospheric scientist chooses the MODIS/VIIRS image through the web-interface, (b) the selected region of interest is used to subset the MODIS/VIIRS image, (c) the subset image is divided into overlapping patches (d) and the patches are archived with the corresponding labels.

box over a subset of a MODIS/VIIRS swath which is associated with the cloud and aerosol categories. To further avoid label

inconsistencies in the training dataset, the following guidelines were followed:



1. The MODIS/VIIRS image subset must only contain the specific cloud and aerosol categories. E.g. a subset image must only contain a collection of cumuliform clouds.

2. Except for cumuliform and cirrus/high-altitude clouds, avoid the inclusion of cloud or aerosol edges wherever possible.

With these guidelines the training dataset excludes examples of cloud and aerosol types that are similar to each other, e.g. cases
are avoided where a cloud that is equally likely to be transitional/mixed clouds and close-stratiform.

Further quality assurance (QA) was conducted on the images that were labeled transitional/mixed and cumuliform clouds, and aerosols. For a chosen small patch size and a subset MODIS/VIIRS image, cumuliform or transitional/mixed clouds could have in-between regions that are considered "clear-air". The MODIS/VIIRS level-2 cloud-mask was used to screen out patches within cumuliform or transitional/mixed subset images that are "clear-air", since for day-time deep-ocean images the
MODIS/VIIRS cloud-mask reliably identifies clouds (Frey et al., 2008)[2].

## 2.3 Overview of CNNs in the context of aerosol and cloud type identification

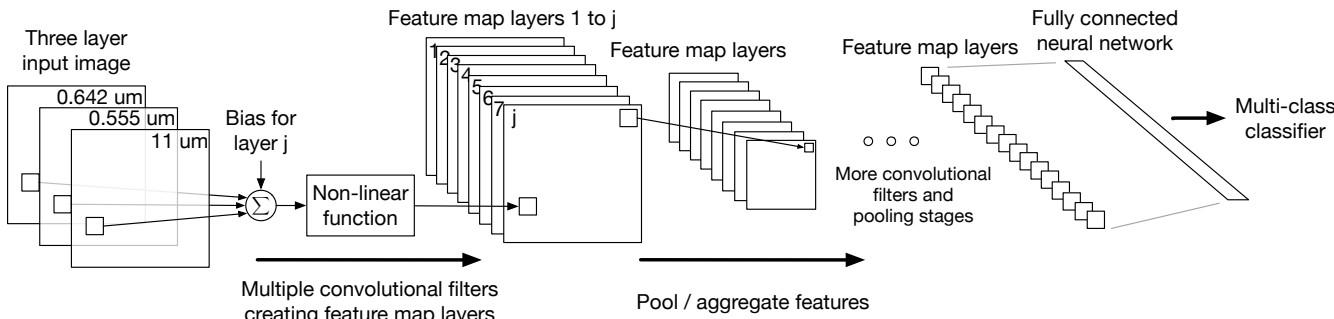

**Figure 5.** A Convolutional Neural Network (CNN) extracts local spatial features from an image and combines the local spatial features to higher order features. The higher order features are then used to linearly separate different image types; this figure shows a simplified diagram of a CNN which gives an idea of how the local spatial features and higher order features are created. In the first CNN layer, after the input image, the CNN has multiple multi-layer convolution filters that produce feature maps: for each multi-layer filter 1) the CNN convolves the filter with the input image, 2) a bias term specific to the feature map is added to each convolution sum, 3) the results are passed through a non-linear activation function and 4) the outputs are then placed in a feature map; the non-linear function with the convolution sums and biases gives the CNN the ability to find the non-linear separation between different image types. The feature maps produced by the first CNN layer contains local extracted spatial features. The next CNN layer converts the local extracted spatial features into higher order features by pooling/aggregating the features in each feature map separately using either an averaging or max operator. The convolution and pooling/aggregation steps are repeated until a series of small feature maps are produced. Depending on the architecture of the CNN, the final layer of feature maps is then passed through a fully connected NN and then finally through a multi-class classifier.

---

[2]The VIIRS and MODIS level-2 cloud-mask use the same algorithm, except for VIIRS less spectral bands are used.



A set of convolution filters can be used to make distinctions between different image types based on their spatial properties. Based on the response of the filters a decision is made in what category (i.e. label) the input image belongs. E.g., if we want to make a distinction between "clear-air" and cumuliform cloud images, the output of a high-pass filter would emphasize spatial variation in a cloudy image.

A CNN essentially consist of a series of convolution filters that are tuned to emphasize various spatial properties in an image; each filter produces a feature map. The feature maps are passed through pooling/aggregation (i.e. down-sampling) operations that allows a CNN to be less sensitive to shift and distortions of the spatial properties in the images (Simonyan and Zisserman, 2014; LeCun et al., 1989a; Krizhevsky et al., 2012; Szegedy et al., 2017; LeCun et al., 1998, 1989b). More specifically, a CNN extracts local spatial features from an image and combine all the local spatial features to higher order features via the pooling/aggregation operations, and the higher order features can be used make a distinction between different image types; Fig.5 shows a simplified diagram of a CNN which gives an idea of how the local spatial features and higher order features are created through the convolutional filters and pooling/aggregation operations.

Similar to fully connected Neural Networks (NNs), the output of the convolution filters are passed through non-linear functions (activation functions) with a bias term; the purpose of the non-linear function with the bias term is to "activate" the output of a convolution filter if the output exceeds some threshold value set by the non-linear function and bias term. E.g., we can emphasize two different levels of variability in an image by connecting a high-pass filter to two ramp functions ($f(x) = \max(0, x + b)$) with different bias values $b$; the ramp function basically acts as a threshold function. Fig. 6 shows an example of the output of a three dimensional (3D) convolutional filter that is produced from a cirrus/high-altitude cloud image. Each layer of the 3D-filter in Fig. 6 convolves with each corresponding layer of the input image; the Discrete Fourier Transforms (DFTs) in Fig. 6 show that the filter layers are low- and band-pass filters. With the bias term and non-linear activation function the 3D-filter produces an image that highlights the local high-reflectivity and "clear-air" regions in the cirrus/high-altitude cloud.

The layers of filters and pooling/aggregation operations with non-linear activation functions, which produce feature maps, are repeated several times. The last layer of the feature maps are passed through a fully connected NN and the output is an one-dimensional vector as shown in Fig. 5. The number of consecutive filter and down-sampling operations (i.e. depth) increases the expressive power of a CNN (i.e. make the CNN more accurate). The one-dimensional output vector of a CNN is typically passed through a multi-class classifier that models a probability value of each label for the input image.

Although fully connected NNs have the ability to non-linearly separate vectors (e.g. images) of different types (Friedman et al., 2001), a fully connected NN is a suboptimal candidate for image recognition tasks since it has no built-in invariance to shifts and distortions in images or structured vectors (LeCun et al., 1998).

Several different CNN configurations have been proposed where each CNN configuration has different convolution dimension sizes, the number of features maps and the depths differ, the sequence of the convolution and pooling/aggregation differ, etc. (Szegedy et al., 2017; Simonyan and Zisserman, 2014). It is not clear which CNN configuration is an optimum choice for a particular image classification problem. In this paper we chose a CNN based on the 1) accuracy ranking of various Ten-





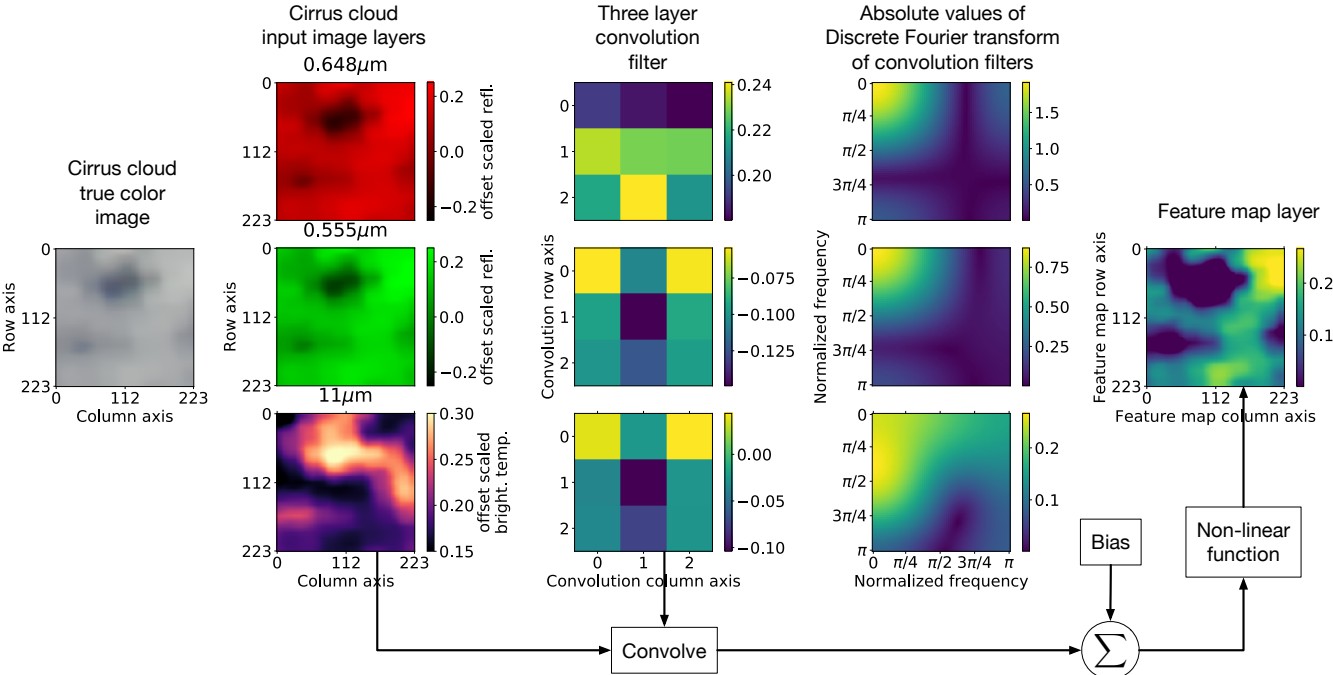

**Figure 6.** This figure shows an example of a feature map that is produced from a cirrus/high-altitude cloud image by a specific 3-dimensional (3D) convolution filter of the first layer of a Convolutional Neural Network (CNN). The layers of the 3D-filter are low- and band-pass filters as indicated by the Discrete Fourier Transforms (DFTs). The 3D convolution filter is convolved over the input image; for each convolution sum a bias term is added and then passed through a non-linear activation function which corresponds to feature map pixel. For this example the feature map highlights the local high-reflectivity and "clear-air" regions in the cirrus/high-altitude cloud.

sorflow CNN implementations (Google, 2020) and 2) accessibility of the Tensorflow CNN implementation (Silberman and Guadarrama, 2016).

## 2.4 Pre-trained and fine-tuned CNNs

A CNN that classifies images typically requires a very large training labeled dataset (e.g. a million labeled images) to estimate
the parameters of the CNN (i.e. the convolutional filters and bias terms), since the number of parameters required to accurately identify different image types increases exponentially with the size of the input images (Gilton et al., 2020). In this paper the training dataset is relatively small since it is time-consuming to hand-label MODIS/VIIRS images and for optically thick aerosol there are not enough events in the observations.

    To address the issue of a small training dataset, we take advantage of a remarkable property of CNNs: if the CNN's convo-
lution filters were tuned (i.e. estimated) on a general set of images (e.g. a wide range of internet image) to make a distinction between image types, the same tuned convolutions filters can be used in the image recognition domain of an entirely different problem. More specifically, the convolution filters in a CNN is kept fixed and the parameters of the multi-class classifier





is re-estimated with a training dataset in the new image recognition domain; we call this type of CNN a pre-trained CNN. E.g., a pre-trained CNN that was trained on internet image has been employed successfully in a radar remote-sensing applica-
tion (Chilson et al., 2019).

The reason why a pre-trained CNN's convolutional filters are "transferable" to a different image recognition domain, is because 1) concepts of edges and different smoothness properties in images are shared across various image domains, 2) convolutional filters by themselves are agnostic to the image recognition domain since they quantify smoothness properties of images and 3) re-estimation of the multi-class classifier parameters adjust for the how the convolution filters respond to the
new image recognition domain.

We demonstrate through our results that the method that employs the pre-trained CNN feature extractor (FE) can provide accurate classification results, though the classification accuracy is dependent on the patch size; furthermore.

Since it is not clear whether a pre-trained CNN is the best CNN configuration to identify cloud types and optically thick aerosol features, in this paper we also consider a fine-tuned CNN. A fine-tuned CNN is created by 1) taking the pre-estimated
convolution filters of a pre-trained CNN as initial values, 2) the initial values are placed in a new CNN and 3) the CNN is trained on the training dataset of MODIS/VIIRS images (Sun et al., 2017). In other words, a fine-tuned CNN is basically a pre-trained CNN that has been adjusted for another image domain.

## 2.5 The feature extraction algorithms

### 2.5.1 Pre-trained CNN feature extractor

FE-Alg. I gives an outline of the pre-trained CNN FE-algorithm. The pre-trained CNN that is used is the Inception-v4 CNN (Szegedy et al., 2017).    In FE-Alg. I the input image is normalized such that each pixel for the average input im-

---

**FE-Algorithm I** `feature_extractor = P-CNN:` pre-trained CNN feature extractor

---

**Input:** Patch $X$ with patch size $S$ at wavelengths 0.642 $\mu$m, 0.555 $\mu$m and 11 $\mu$m

  1: Normalize the input image.

  2: Upsample image to size 299 by 299 pixels with bi-linear interpolation

  3: $Z =$ apply `Inception-v4` pre-trained convolutional neural network

**Assume:** The feature vector $Z$

---

age has zero mean and a standard deviation of one. The Inception-v4 CNN was trained on normalized images to improve performance (Orr and Müller, 2003, Chp. 1). The coefficients that are used to normalize the input image are computed from the training dataset. The Inception-v4 CNN takes as input images with size 299 by 299 pixels; hence the MODIS/VIIRS image
patches are upsampled using bi-linear interpolation.





### 2.5.2 Fine-tuned CNN feature extractor

FE-Alg. II gives an outline of the fine-tuned CNN FE. The pre-trained VGG-16 CNN was chosen as the basis of the fine-tuned CNN rather than a pre-trained Inception-v4, since VGG-16 has fewer parameters (i.e. degrees of freedom) compared to that of the Inception-v4 CNN. The parameters of the fine-tuned VGG-16 CNN are estimated separately from Alg. 1 with the same

---

**FE-Algorithm II** `feature_extractor = F-CNN`: pre-trained CNN feature extractor that has been fine-tuned

---

**Input:** Patch $X$ with patch size $S$ at wavelengths 0.642 $\mu$m, 0.555 $\mu$m and 11 $\mu$m

1: Normalize the input image.

2: Upsample image to size 224 by 224 pixels with bi-linear interpolation

3: $Z =$ apply `VGG16` fine-tuned pre-trained convolutional neural network

**Assume:** The feature vector $Z$

---

training dataset that is used by Alg. 1; the fine-tuning is accomplished using a software package such as Tensorflow (Abadi et al., 2016). As with the Inception-v4 CNN the input image to the fine-tuned VGG-16 CNN is normalized. The VGG-16 CNN takes as input images with size 224 by 224 pixels; hence the MODIS/VIIRS image patches are upsampled using bi-linear interpolation.

### 2.5.3 Mean-standard deviation feature extractor

FE-Alg. III gives an outline of the mean and standard deviation FE; this algorithm computes the mean and standard deviation

---

**FE-Algorithm III** `feature_extractor = MeanStd`: Mean and standard deviation feature extractor

---

**Input:** Patch $X$ with patch size $S$ at wavelengths 0.642 $\mu$m, 0.555 $\mu$m and 11 $\mu$m

1: $Z =$ compute mean and standard deviation at each wavelength

**Assume:** The feature vector $Z$

---


(SD) for each input wavelength, and the mean and SD values are placed in a six element vector.

## 3 Results - Case studies and sensitivity analysis

Table 2 shows the shorthand notation of the methods that were used to produce the results; recall that each method, which is created by Alg. 1, consists of a multi-class classifier with a FE-algorithm and Alg. 2 applies the method on MODIS/VIIRS

images. E.g., the shorthand notation `P-CNN`$^{Appp}$ indicates that the method's multi-class classifier is paired with pre-trained CNN FE-algorithm (i.e. FE-Alg. I) which operates on a patch size of $ppp$ by $ppp$ pixels.

In our results we seek to answer the following questions:

1. Can our methodology reliably identify optically thick aerosol which the MODIS/VIIRS level-2 products struggle to detect?


| Short hand notation | FE algorithm | Description of FE |
|---|---|---|
| P−CNN$^{Appp}$ | FE-Alg. I | With pre-trained CNN |
| F−CNN$^{Appp}$ | FE-Alg. II | With fine-tuned CNN |
| MeanStd$^{Appp}$ | FE-Alg. III | With baseline mean-standard deviation FE |

**Table 2.** Method mnemonics that are used throughout this result section and the corresponding feature extraction (FE) algorithms.

2. Can our methodology make a distinction between different cloud types?

3. How well does our methodology label MODIS/VIIRS images that are not part of the training and test datasets? In other words, how well does our methodology generalize?

With the first two questions we qualitatively validate our results with four case studies, since we are unaware of a quantitative dataset to characterize transitional/mixed, closed-stratiform and cumuliform clouds and any dataset that list aerosol events that the MODIS/VIIRS level-2 products fail to detect. We answer the third question by quantifying cirrus/high-altitude cloud and aerosol identification accuracies as a function of optical depth (OD) by using CALIOP cloud OD and both MODIS and CALIOP aerosol OD (AOD) measurements. Consequently, since we use CALIOP OD measurement to validate our results we only used Aqua-MODIS observations because SNNP-VIIRS is not consistently in the same orbit as CALIOP.

### 3.1 Training and test datasets, and test errors

Recall that two labeled datasets with patch sizes 100 and 25 pixels were created from the adapted NASA Worldview website (see Fig. 4 and Sect. 2.2), and from each labeled dataset training and validation datasets were assembled. Table 3 shows the number of patches in the training and test datasets per patch size, and number of unique MODIS/VIIRS image subsets from which the patches were extracted; the quantities in the second column is less than the third because the some images subsets were smaller than 100 by 100 pixels.

Table 4 show the test errors of the different methods. The test error is defined as the average number of labeled patches that are misclassified from the test datasets, where the estimated parameters of the methods are independent from the patches in the test datasets. Although the test error is informative in comparing the classification performances of the different methods (Friedman et al., 2001), the test error is not necessarily an accurate predictor of the out-of-sample (i.e. samples that are not in the training and test datasets) misclassification error (Recht et al., 2019).

Table 4 clearly shows that the method with the fine-tuned CNN (i.e. F−CNN$^{A025}$) has a better error performance compared to method with a pre-trained CNN (i.e. P−CNN$^{A025}$).





| Number of ... | 100 by 100 pixels | 25 by 25 pixels |
|---|---|---|
| Training patches | 50 283 | 148 284 |
| Test patches | 16 761 | 49 428 |
| "Clear-air" scenes | 46 | 142 |
| Aerosol scenes | 148 | 343 |
| Cumuliform scenes | 38 | 67 |
| Transitional/mixed scenes | 20 | 35 |
| Closed-stratiform scenes | 31 | 47 |
| Cirrus/high-altitude scenes | 102 | 119 |

**Table 3.** The first two rows shot the number of patches in the training and test datasets per patch size. The rest of the rows show the number of unique MODIS/VIIRS image subsets from which the patches were extracted.

| | 25 by 25 pixels | | | 100 by 100 pixels | |
|---|---|---|---|---|---|
| FE-Algorithm | F-CNN$^{A025}$ | P-CNN$^{A025}$ | MeanStd$^{A025}$ | P-CNN$^{A100}$ | MeanStd$^{A100}$ |
| Test error | 0.11% | 4.84% | 4.08% | 0.45% | 0.38% |

**Table 4.** The test errors of the different methods with different patch sizes. For patch size 25 by 25 pixels the method with the fine-tuned CNN FE-algorithm F-CNN$^{A025}$ has the smallest error. For patch size 100 by 100 pixels the mean standard-deviation FE-algorithm MeanStd$^{A100}$ performs better than the pre-trained CNN FE-algorithm P-CNN$^{A100}$. The case studies and sensitivity analyses results, shown elsewhere, demonstrate that the F-CNN$^{A025}$ identifying optically thick aerosol and cirrus clouds at higher accuracies compared to the other methods. The discrepancy between the test errors and what are observed in case studies and sensitivity results indicate that test error is not necessarily an accurate predictor of the out-of-sample (i.e. samples that are not in the training and test datasets) misclassification error.

It should be noted that unlike the patches in the training and test datasets that do not contain a mixture of cloud types and aerosol (see Sect. 2.2), patches in MODIS/VIIRS imagery often have mixture of cloud types. For a patch that contains a mixture of cloud types and/or aerosol, we expect that the methods will label a patch that has the most dominant cloud type or aerosol.

## 3.2 Case studies

Fig. 7, 8, 9 and 10 show four MODIS-Aqua scenes with the labeled results of four methods; we do not show the result of method P-CNN$^{A025}$ since it has the largest test error. Included in the figures are the true color, 11 $\mu$m brightness temperature (BT), MODIS MYD04 AOD and MODIS MYD35 cloud-mask products (Levy et al., 2013; Frey et al., 2008). The scenes in Fig. 7 and 8 were not part of the training or test datasets, and for all images any sun-glint and landmasses have been masked out.
The four MODIS-Aqua scenes were specifically chosen because the MYD04 aerosol product has retrievals which are flagged



as bad by the MYD04 Quality Control (QC) and the MYD35 cloud product labeled the aerosols as clouds (Levy et al., 2013; Frey et al., 2008).

Box A in Fig. 7, 8, 9 and 10 specifically points out where the CNN methods (i.e. F-CNN$^{A025}$ and P-CNN$^{A100}$) were able to correctly label the aerosols plumes and the corresponding MYD04 and MYD35 products have bad QC AOD retrievals
and the aerosols are labeled as clouds; the CNN methods use the spatial texture differences between aerosols clouds to make a distinction which enables them to correctly label these optically thick aerosol, where the fine-tuned CNN method is able to identify aerosols at a finer image resolution. In contrast, the baseline methods (i.e. MeanStd$^{A025}$ and MeanStd$^{A100}$) misclassify aerosols as clouds in box B in Fig. 7, 8, 9 and 10 because the baseline methods use simple statistics which do not captures the complexity of the spatial texture features of aerosols and cloud types.

The fine-tuned CNN method does misclassify some of the edges of aerosols plumes as cumuliform and cirrus/high-altitude clouds. We did not explicitly model the labeling of the edges of aerosols and clouds, and we consider it part of our future work to improve the labeling of aerosol plume edges.

Box C in Fig. 7 and 8 shows the presence of sparse cirrus/high-altitude clouds where the 11 $\mu$m BTs ranges between 249 K[elvin] and 257 K, and in Fig. 9 box C show dense cirrus/high-altitude clouds that cover a larger area with 11 $\mu$m
BTs between 242 K and 253 K. For the cases where the cirrus/high-altitude clouds are more sparse as in Fig. 7 and 8 the methods with patch size 25 pixels are more sensitive to detecting the cirrus/high-altitude clouds compared to the methods with patch size 100 pixels. In the following sub-section we quantify the how sensitive the different methods are in detecting the cirrus/high-altitude clouds.

Though for closed-stratiform, transitional/mixed and cumuliform clouds there is no clear approach to objectively validate
the results of the various methods, we point out where these clouds were detected in the case studies with the true color imagery providing validation. Box D in Fig. 8 and 9 show possible closed-stratiform clouds which are near or under smoke plumes. In Fig. 8 the fine-tuned CNN method was able to detect the closed-stratiform cloud south of the smoke plume. In Fig. 9 box D.1 shows that a portion of the closed-stratiform is under the smoke plume between, and box D.2 shows another closed-stratiform cloud sandwiched between a cirrus/high-altitude cloud and the smoke plume. For box D.1 methods P-CNN$^{A100}$ and
MeanStd$^{A100}$ identify a larger portion of the closed-stratiform cloud compared to F-CNN$^{A025}$ and MeanStd$^{A025}$, since the 25 pixel methods can resolve the closed-stratiform at a higher image resolution which is also evident in box D.2; the baseline methods incorrectly identifies more closed-stratiform clouds compared to the other methods.

Box E in Fig. 8 show where the methods separate cumuliform from transitional/mixed clouds.

Box F in Fig. 10 shows a part of the clouds that were labeled as cirrus/high-altitude by the fine-tuned CNN method where
the 11 $\mu$m BT is more than 278 K. A possible reason for this misclassification is because the cirrus/high-altitude labeled observations in the training dataset are contaminated with other cloud types.

### 3.3 Generalization study through cirrus/high-altitude and aerosol sensitivity analyses

To understand how well our methodology generalizes in labeling MODIS/VIIRS observations that were not part of the training and test datasets, we generated identification accuracy statistics using both the MODIS MYD04 and CALIOP aerosol products





and the CALIOP cloud product. Given that CALIOP spatial footprint is 90 meters, only the patches that intersects with the CALIOP footprint were processed by the methods.

### 3.3.1  Cirrus/high-altitude cloud sensitivity analysis

Fig. 11 shows the average fraction of cirrus/high-altitude clouds, per OD interval, detected in the intersection of a CALIOP footprint and patch by the different methods and the MODIS cloud mask (CM) where the CALIOP cloud product indicated that

*only* ice-clouds were present. The reported fraction of cirrus/high-altitude cloud statistics is from all of July 2016; CALIOP observations were excluded from the analysis that had water cloud and aerosol observations present at any altitude. The MODIS cloud mask in Fig. 11a and c provides a benchmark for detecting clouds.

Fig. 11a and c show the fraction of cirrus/high-altitude clouds detected by the methods operating on 100 and 25 pixel patches, respectively; the fine-tuned CNN method detects cirrus/high-altitude clouds at a higher accuracy compared all the other

methods. Fig. 11b and d give insight as to why accuracy of detecting cirrus/high-altitude clouds decreases as the cirrus/high-altitude cloud OD decreases. Fig. 11b shows that for OD less than three cirrus/high-altitude clouds are typically labeled as cumuliform clouds by the pre-trained CNN method. The cumuliform labeling of cirrus/high-altitude clouds should not be regarded as a consistent misclassification, since tenuous cirrus/high-altitude clouds over the ocean are commonly surrounded by cumuliform clouds in a patch size of 100 pixels. Once the patch size decreases to 25 pixels Fig. 11d shows that the fine-

tuned CNN method identifies tenuous cirrus/high-altitude clouds with OD of less than one, and are misclassified as "clear-air", though cumuliform clouds are still present for cirrus/high-altitude clouds with ODs up to three.

### 3.3.2  Aerosol sensitivity analysis

Fig. 12 shows the average fraction of aerosols, per AOD interval, detected in the intersection of a CALIOP footprint and patch by the different methods where the MODIS MYD04 or CALIOP aerosol products indicated that aerosols were present

in the complete intersection. The reported fraction of aerosols statistics are from March 2012 up to May 2018, and aerosol observations were excluded from the analysis if 1) any cloud was present according to both the MODIS and CALIOP cloud products and 2) if the confidence in the aerosol AOD retrieval was less than "high-confidence" as indicated by the MYD04 QC.

In Fig. 12 the insensitivity to detecting tenuous aerosols by the CNN and baseline methods is by design, since the aerosol

training data was manually generated from true color imagery which is not sensitive to thin aerosol. Fig. 13 shows what the AOD of the aerosol patches are in the training dataset through the MYD04 AOD measurements; the median AOD of the aerosol patches are larger than 0.6 for both patch sizes.

Fig. 12 show that the sensitivity to detecting aerosol decrease as the patch size decreases, which is an opposite trend compared to the cirrus cloud identification accuracy versus patch size. Fig. 13 gives a possible reason why there is a decrease in sensitivity:

the AOD distribution of aerosol patches overlap more with "clear-air" patches in the 25 pixel patch size training dataset compared to the larger patch size training dataset.





According to Fig. 12a and c the MODIS aerosol product of the proposed methodology can accurately identify optically thick aerosol. But, the CALIOP aerosol product indicates otherwise in Fig. 12b and d. The reason why the proposed methodology's aerosol identification sensitivities relative to MODIS (MYD04) and CALIOP differ could be a discrepancy between the MODIS and CALIOP measured AODs as shown in the two-dimensional histogram in Fig. 12e. E.g., for a fixed CALIOP AOD of 2 the corresponding MODIS AOD ranges between 0.25 and 1; in other words, what might be considered optically thick by the CALIOP, could be considered optically thin by the MODIS aerosol product. It is unclear why there is such a large disagreement between the MODIS and CALIOP AODs.

## 4 Conclusion

### 4.1 Summary

We introduced a new methodology that identifies distinct cloud types and optically thick aerosol by leveraging spatial textual features from radiometer images that are extracted via CNNs. We demonstrated through several case studies and via comparisons of the MODIS and CALIOP cloud and aerosol products:

1. The proposed methodology can identify low-mid level cumuliform, closed-stratiform, transitional/mixed (transition between the two or mixed types), or cirrus/high-altitude cloud types; these cloud regimes are related to meteorological conditions due to the strong correlations between cloud types and atmospheric dynamics (Levy et al., 2013; Tselioudis et al., 2013; Evan et al., 2006; Jakob et al., 2005; Holz, 2002).

2. The cirrus/high-cloud type cloud detection accuracy is a strong function of the image patch size, where a smaller patch size yield higher accuracy. We found that cirrus/ high-clouds are often misclassified as "clear-air" and cumuliform clouds resulting from the frequent cumuliform clouds beneath tenuous cirrus clouds.

3. The proposed methodology can reliably identify these optically thick aerosol events, whereas the MODIS level-2 aerosol product labeled the AOD retrievals of the optically thick aerosol as "bad" with the quality control flags (Levy et al., 2013). The implication of relying on the MODIS level-2 aerosol product quality control in a climatological research project involving aerosols is that most the large impact optically thick aerosol will be excluded in the research study; this problem stresses the importance for the reliable identification of optically thick aerosol.

4. Optically thick aerosol detection is also a strong function of the image patch size, where smaller patch sizes yield lower accuracy. The insensitivity to tenuous aerosols is by design, since the human labeled training dataset only contained optically thick aerosols. For future work we consider increasing the number of lower AOD aerosol labeled images in the labeled dataset in order to increase the aerosol detection sensitivity.

Unfortunately, since we do not know of any dataset of labeled cumuliform, closed-stratiform, transitional/mixed clouds, there was no clear approach to quantify the detection accuracy of these clouds; we leave it to future work to characterize this accuracy performance.





A surprising aspect of our methodology is that the CNNs that we employ are pre-trained on non-radiometer images (e.g. random images from the internet). In more detail, the convolutional filters of the CNNs which extract the spatial textual features from image were estimated from non-radiometer images; employing pre-trained CNNs in other remote sensing problems with promising results have been recently reported in the literature (Chilson et al., 2019). We demonstrated in this paper that with additional fine-tuning of a pre-trained CNN, by further estimating the convolutional filters where the initial values are of a pre-trained CNN, the classification performance of radiometer images can be further improved upon.

## 4.2 Future work

Our proposed methodology is considered preliminary work and further work will be required to maximize the methodology's operation and scientific use. The current limitations are that 1) the methodology is computationally inefficient, 2) cloud type and aerosols are only identified over deep-ocean images and 3) the methodology only operates on daytime observations. Currently a total of 20 minutes is required to label a MODIS 5-minute granule with a GPU accelerated fine-tuned CNN method. This computational inefficiency is because each image patch is processed independently from the other patches although the patches overlap. Consequently, the CNN convolutional operations on each pixel are repeated several times. This computational redundancy can be eliminated by adapting the CNN architecture and potential computational speed can be increased at least by a factor of five.

We consider the deep-ocean and daytime only limitations as future research directions. Cloud type and aerosol identification over land is a much harder research problem since the earth surface is non-uniform and changes over time; in our future work we plan to model the earth surface and incorporate the land-surface model in our CNN methodology. In regard to the daytime only limitation, the CNN methods in the paper have only been applied on three spectral channels ($0.642\mu$m, $0.555\mu$m and $11\mu$m) and it would be interesting to understand how the cloud type and aerosol identification accuracies change when some spectral bands are removed or added from the input images. E.g., how would the CNN identification accuracies be impacted if only infrared radiance measurements are used?

Another future research direction is how to adapt our methodology to Geostationary Earth Orbiting (GEO) imagers? GEO imagers provide temporal multi-spectral image sequences where the temporal resolution is at least 10 minutes. A compelling research question would be: How can we leverage the temporal component of the GEO images to identify cloud types and aerosols while incorporating cloud and aerosol physical models with the machine learning methodology? The temporal information content will allow the machine learning to separate high- from low-altitude clouds, while the cloud and aerosol physical models will provide additional information that can constrain the cloud type and aerosol identification.

*Code availability.* Publicly available software were used to produce the results in this paper. `Tensorflow` was used to for the Convolutional Neural Networks (CNNs), and the pre-trained models are available at (Google, 2020). `scikit-learn` was used to model the multinomial classifier (Pedregosa et al., 2011).



*Data availability.* All data used in this study are publicly available. The MODIS and VIIRS data are available through the NASA Level-1
and Atmosphere Archive and Distribution System Web Interface (https://ladsweb.modaps.eosdis.nasa.gov/, NASA 2020). The CALIOP data
were obtained from the NASA Langley Research Center Atmospheric Science Data Center (https://eosweb.larc.nasa.gov/project/calipso/
calipso_table, NASA Langley Research Center, 2020).

*Author contributions.* Authors RMW and WJM developed the statistical image processing methodologies. Author JSR created the labeled
datasets. Author REH provided expert advice on MODIS, VIIRS and CALIOP datasets. All authors participated in the writing of the
manuscript.

*Competing interests.* The authors declare that they have no conflict of interest.

*Financial support.* This research has been supported by ONR-322 N00173-19-1-G005, DOE DE-AC02-06CH11357, NSF OAC-1934637
and NSF DMS-1930049.

*Acknowledgements.* We would like to thank Davis Gilton for his thoughtful suggestions on fine-tuning CNNs.



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

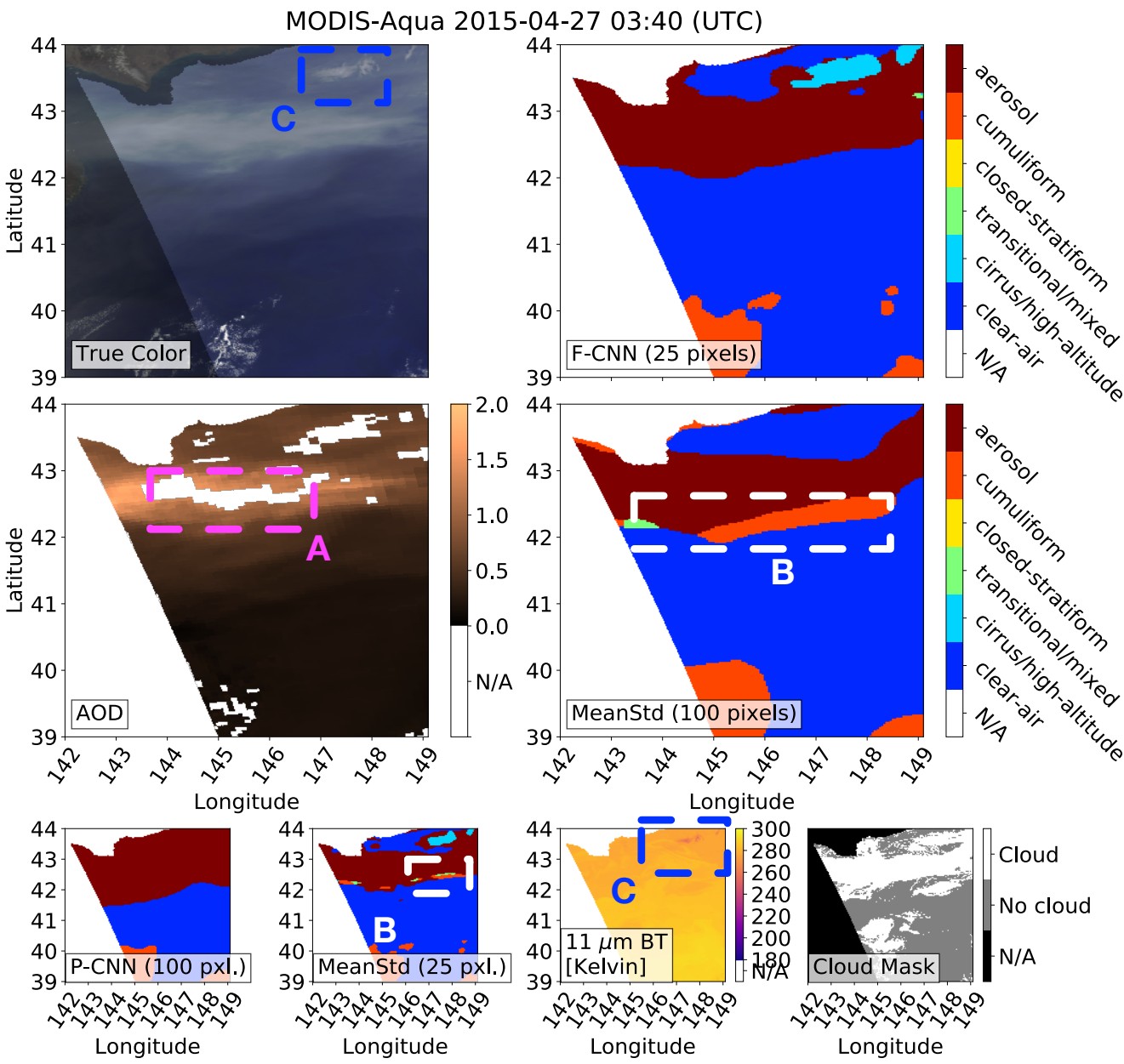

**Figure 7.** Box A show where the 1) MODIS MYD04 aerosol product has aerosol optical depth (AOD) retrievals which are flagged as bad by the MYD04 quality control, 2) the MODIS MYD35 cloud-mask product labeled the aerosols as clouds, 3) the pre-trained 100 pixel CNN and fine-tuned 25 pixel CNN (F-CNN) methods were able to successfully detect the extreme aerosols. Box B show where the baseline mean-standard deviation (MeanStd) method misclassified the aerosols as clouds. Box C show the presence of sparse cirrus/high-altitude clouds with a 11 $\mu$m brightness temperatures (BTs) ranges between 249 K[elvin] and 257 K, and the 25 pixels methods (F-CNN and MeanStd) were able to detect the cirrus/high-altitude cloud. For all images any sun-glint and landmasses have been masked out.



**Figure 8.** Box A show where the 1) MODIS MYD04 aerosol product has aerosol optical depth (AOD) retrievals which are flagged as bad by the MYD04 quality control, 2) the MODIS MYD35 cloud-mask product labeled the aerosols as clouds, 3) the pre-trained 100 pixel CNN and fine-tuned 25 pixel CNN (F-CNN) methods were able to successfully detect the extreme aerosols. Box B show where the baseline mean-standard deviation (MeanStd) method misclassified the aerosols as clouds. Box C show the presence of sparse cirrus/high-altitude clouds with a 11 $\mu$m brightness temperatures (BTs) ranges between 249 K[elvin] and 257 K, and the 25 pixels methods (F-CNN and MeanStd) were able to detect the cirrus/high-altitude cloud. Box D show possible closed-stratiform clouds which are near the smoke plume that was detected by the F-CNN method. Box E show where the F-CNN method separates cumuliform from transitional/mixed clouds. For all images any sun-glint and landmasses have been masked out.

**Figure 9.** Box A show where the 1) MODIS MYD04 aerosol product has bad aerosol optical depth (AOD) retrievals which are flagged as bad by the MYD04 quality control, 2) the MODIS MYD35 cloud-mask product labeled the aerosols as clouds, 3) the pre-trained 100 pixel CNN and fine-tuned 25 pixel CNN (F-CNN) methods were able to successfully detect the extreme aerosols. Box B show where the baseline mean-standard deviation (MeanStd) methods misclassified the aerosols as clouds. Box C show the presence of dense cirrus/high-altitude clouds with a 11 $\mu$m brightness temperatures (BTs) ranges between 242 K[elvin] and 253 K, and all four methods were able to detect the cirrus/high-altitude cloud. Box D show possible closed-stratiform clouds which are near or under smoke plume. All four methods were able to detect the closed-stratiform clouds to various degrees, whereas the 25 pixel methods are able to detect the closed-stratiform clouds at -127 longitude at a finer image resolution compared to the 100 pixel methods. For all images any sun-glint and landmasses have been masked out.

**Figure 10.** Box A show where the 1) MODIS MYD04 aerosol product has bad aerosol optical depth (AOD) retrievals which are flagged as bad by the MYD04 quality control, 2) the MODIS MYD35 cloud-mask product labeled the aerosols as clouds, 3) the pre-trained 100 pixel CNN and fine-tuned 25 pixel CNN (F-CNN) methods were able to successfully detect the extreme aerosols. Box B show where the baseline mean-standard deviation (MeanStd) methods misclassified the aerosols as clouds. Box F shows a part of the clouds was misclassified as cirrus/high-altitude by the F-CNN method, which is possibly due to cirrus/high-altitude labeled observations in the training dataset that has some contamination of other cloud classes. For all images any sun-glint and landmasses have been masked out.

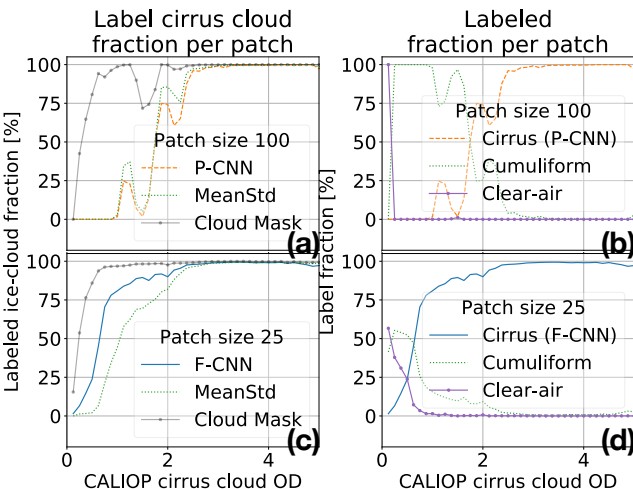

**Figure 11.** For the patches that intersect with the CALIPSO satellite track and the CALIOP cloud product indicated that *only* ice was present, (a) to (d) show the fraction of cirrus/high-altitude clouds detected versus cloud optical depth (OD) in the intersection by the different methods of the proposed methodology and the MODIS cloud mask (CM). (a & b) and (c & d) show the fraction of cirrus/high-altitude clouds detected for methods operating on 100 and 25 pixel patches, respectively. (b & d) show what the patch intersections were labeled as, versus optical depth, by the pre-trained convolutional neural network (P-CNN) and fine-tuned CNN (F-CNN) methods. The F-CNN method detects cirrus/high-altitude clouds at a higher accuracy compared to the other methods at different patch sizes. For both patch sizes of 100 and 25 pixels, the misclassifications are due to cumuliform clouds, since tenuous cirrus/high-altitude clouds are frequently above by cumuliform clouds. At patch size 25 pixels, tenuous cirrus/high-altitude clouds are also prone to be misclassification as "clear-air" as expected. The MODIS cloud mask in (a & c) provides a benchmark for detecting clouds.



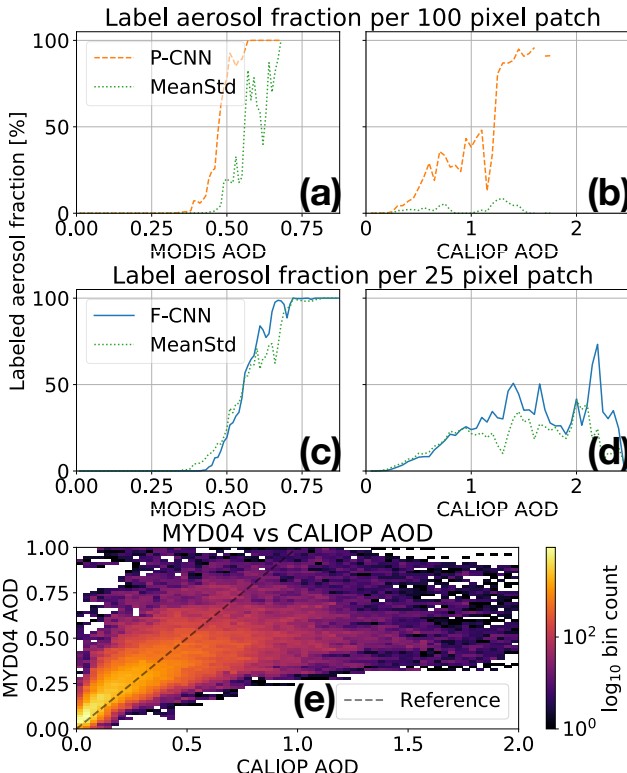

**Figure 12.** (a) to (d) show the average fraction of aerosols detected in the intersection by the different methods of the proposed methodology for the patches that intersect with the CALIPSO satellite track, and either the MODIS MYD04 or CALIOP aerosol products indicated that there were aerosol present in the complete intersection without any clouds. (a & c) show the aerosol fraction versus MODIS MYD04 aerosol optical depth (AOD) and (b & d) show the aerosols fraction versus CALIOP AOD, where (a & b) and (c & d) are specific to the methods of patch sizes 100 and 25 pixels, respectively. The insensitivity to detecting tenuous aerosols by the convolutional neural network (CNN) and Mean-Standard Deviation (MeanStd) methods is by design, since the training dataset intentionally mostly contained optically thick aerosols. Comparing (a & b) and (c & d) indicates that methods operating on 100 pixel patches can detect aerosols with smaller AOD more accurately than methods operating on 25 pixel patches, where the method `MeanStd` can identify aerosols at a slightly lower AOD compared to pre-trained (`P-CNN`) and fine-tuned CNN (`F-CNN`); aerosols that are not identified as typically labeled as "clear-air". (e) gives insight as to why the aerosol identification sensitivities relative to MODIS (a & c) and CALIOP (b & d) differ; e.g. for a MYD04 AOD of 0.5, the corresponding CALIOP AOD ranges between 0.25 and 1.25.



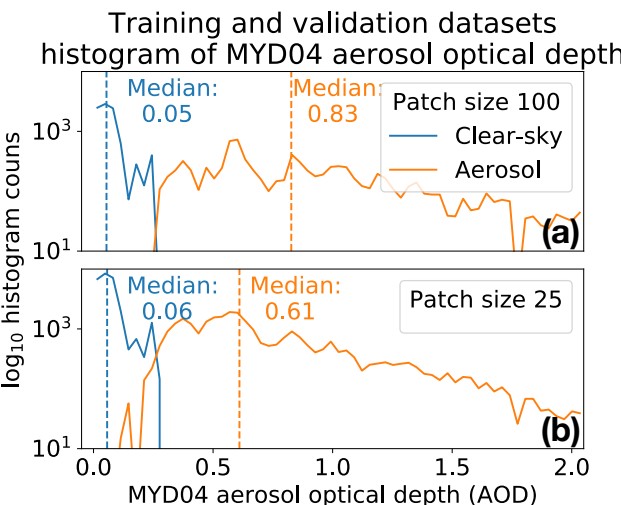

**Figure 13.** (a & b) show the median MODIS MYD04 aerosol optical depth per patch histograms of the training and test datasets of patch sizes 100 and 25 pixels, specifically for the "clear-air" and aerosol labels. There is a clearer AOD separation between the "clear-air" and aerosol labels for the (a) 100 pixel patch size datasets compared to that of the (b) 25 pixel patch size.