# Peer review of "Leveraging spatial textures, through machine learning, to identify aerosol and distinct cloud types from multispectral observations"

_Atmospheric Measurement Techniques, 2020_

## Referee Comment (RC1) · Anonymous Referee #2 · 27 Apr 2020

Marais et al use a machine learning technique together with a new methodology to create a training data set with which they can show that improvements in the discrimination between clouds and aerosols can be achieved. Additionally, these method allows an enhanced capability to classify cloud types.

Before publication in AMT the following minor to major comments should be considered:

[Figure]

**General comments:**

- The introduction is somewhat confusing. It's a mixture between an introduction and a method section and thus needs some restructuring, See my specific comments below.

- Figures captions are quite lengthy and already include an interpretation of the figures which however rather belongs to the main text.

- Several sentences are beginning with "E.g" which in my opinion is not a good writing style.

- Data from several satellite instruments are used in this study. However, these instruments are nowhere in the manuscript described. A short description should be added for each instrument used in this study.

**Specific comments:**

P1, L20: Change "Introduction & Problem Solving" to "Introduction".

P2, L53: References? Who has developed the CNN method?

P3, Fig. 1 and P4, Fig. 2: Both figures should be moved to the method section.

P3, L64: Skip Sub-header "1.1. Objectives".

P5, Section 1.2: Skip section header and move most parts of the text to the method sections. In the introduction it is sufficient to just write in a few sentences what

has been done without going into detail.

P5, L1.3: Skip sub header "1.3 Outline".

P6, Section 1.4: This section should be either included in the method section (extra subsection for this is not necessary). Even better would it be to have this Table in an appendix.

P6, L114: Add a reference for the FE method?

P7, L134: "…….....with which rank the different methods……..". Sentence not correct, something is missing here. Did you mean …." with which "we" rank the different methods"?

P12, L202: The listing is obsolete and mentioning the person to do it is a bit weird here. Therefore, I would suggest to rewrite the sentence as follows ".…...MODIS/VIIRS images that are chosen are divided into overlapping patches……."

P12, L205: "one co-author" obsolete. Simply write "In order to ensure consistent labels for the study, a labeled data set was created…….."

P12, L207: Why are quotation marks used for clear air?

P12, Section 2.2.2: This section sounds like a user manual. Avoid writing the atmospheric scientist/user. Just describe the process itself.

P13, Figure 4 caption: Rephrase sentence to " The web-interface through which a labeled data set of aerosols and cloud types can be created."
P13, Figure 4 caption: Skip the sentence "The user has the option. . ... ..."

P13, L225: ". ...the following guidelines were followed:" Rephrase sentence to avoid repetition of "follow".

P15, L267: Use a full stop after proposed and add the references here. Continue then with "Thereby, each CNN configuration has e. g. different convolution dimension sizes, different number of feature maps, different depths, different sequence of convolution and different pooling/aggregation.

P17, L292: Delete furthermore or move it at the begin of the sentence.

P19, L334: Delete the first "and" in the Section header so that it reads "Training, test datasets, and test errors".

P22, L416: Sentence on Fig13 not clear. Please rephrase.

P23, L428: Are studies existing that compare these instruments and/or document these differences? If yes, please add these references.

P23, L447: By design of what? The CNN?

P33, Figure 11 caption: Some text here belongs rather to the main text since it is describing what conclusions can be drawn from the figures and not what actually is shown.

P34, Figure 12 caption: Same here. Shorten the caption text and just write what is shown.

**Technical corrections:**

P15, L269: optimum choice → best choice

P19, L338: the some → some or the same?

P20, L351: methods or cases?

P19, L346: to method → to a method

P21, L360: aerosols → aerosols

P21, L365: aerosols → aerosols

P21, L369: K[elvin] → K

P21, L372: delete "the" before "how"

P21, L374: Through for → For

P22, L3399: compared → compared to

P22, L418: show → shows

P23, L433: add "that" or "the following" at the end of the sentence.

P23, L443: most the → most of the

P30, Figure 8 caption, L3,4,6,7: show → shows

[Figure]

P30, Figure 8 caption: K[elvin] → K

P30, Figure 8 caption: range → "in the range" or "ranging"
* * *

---

## Short Comment (SC1) · 18 May 2020

The research is very interesting and important. I have some questions and comments regarding the paper.

1. What is the true positive rate and false positive rate of the ML results vs. CALIOP's results in terms of aerosols vs clouds vs clear? The test error provide a mixed information that is hard to interpret.

2. The idea of which is the best patch size captures the aerosol/cloud spatial informa-

tion is very useful. A comparisons between 100 pixels vs 25 pixels are a beginning but not enough to proof that 25 is the best, especially when determine edges of a feature. Is there a sensitivity study to show different patch size?

3. Figure 11 and 12 are kind of confusing. I don't know if I understand it completely, because the optical depth increase not necessarily means that fraction of aerosol increased in my opinion.

4. In the paper there are very detailed descriptions of CNN package including FE and classifier. I think it is hard to separate what is your research vs. what is already built in the CNN packages. When I read the article, I have a feeling I was walked through a detailed introduction of how CNN/deep learning model works. This is maybe because the author assumes audience doesn't have any knowledge of CNN, but I felt like these amount of details make the paper overwhelming.
* * *

---

## Referee Comment (RC2) · Anonymous Referee #1 · 20 May 2020

General comments

This very informative paper presents a novel solution to the outstanding problem of differentiating thin cloud from thick aerosol features in satellite datasets derived from visible imagers. A machine learning (ML) approach based on Convolutional Neural Network (CNN) is presented and discussed. The training datasets are derived from images that are not from radiometers, but are random images from the internet for the initial pre-training and from True Color images classified by hand for the fine tuning of the training. The latter dataset is also used for verification. The reason why this works

is that the algorithm operates based on spatial smoothness of the features and not necessarily on the fact that the images represent radiometric information per se. This is a point of strength of the ML approaches. A series of test cases is presented.

Results show that the the CCN algorithms (bot the fine-tuned with 25x25 pixels and the pre-trained with 100x100 pixels) performed better than the standard deviation method in correctly classifying cloud versus aerosol features. The algorithms do not perform well for the cases of cloud or aerosol egdes, for thin cirrus with optical depth lower than three and thin aerosol cases. The fact that the algorithm does not perform well in thin aerosol situations which were purposefully not included in the human-labelled training dataset highlights the importance of a good design of the training datasets. Any "biases" present in the training datasets will also reflect in the output of the ML algorithms which are basically ignorant of the physics of the problem and only deal with certain characteristics of the image itself.

Overall the paper is well written, robust, well documented with plenty of references and the research presented is of high interest for the community. I have some minor comments below. I would also recommend a bit of rewriting of the algorithm description which I found difficult to follow. I encourage the authors to pursue further this work to improve on the current findings and be able to address the remaining "sticky" points.

Minor comments/typos

line 80: can be labelled

line 95: correctly identify

line 185 :I do not see L and K defined. At this point of the paper, the reader gets a bit lost in the maths (at least I did). Please see if you can keep it more discursive and easier to read. I would suggest putting the mathematical details of the algorithms in an appendix.

line 218: I think this is a really important point. If there are any biases in the training

datasets the algorithms will likely reproduce those biases.

line 223: It might be rather arbitrary to recognize clouds at different altitudes from the True Color images. Could you comment on that. Further down, figure 6 makes this point: how can you be sure that that's a cirrus cloud?

line 283: there's a period in an odd place, should be a comma

line 289: for how

line 292: unfinished sentence

Section 2.5 is difficult to read. Some details could perhaps be given in an appendix and the text in the paper made more fluid.

Section 2.5.3 is a bit too concise. Please expand.

Line 365: yes, the edges are problematic. Please spend a few words in saying how you will address this problem in future work

Line 376: this is a difficult case even for a human-based recognition.

Line 423. "However" instead of "But" to start the sentence

Line 428. This is not the focus of this paper, but are there other studies confirming the inconsistency that you see between MODIS and CALIOP data?

Line 447: this also highlights that it is really important to choose well the training dataset.

Line 450: this is indeed a sticky point.

Line 476. You do not need the question mark at the end.

---

## Short Comment (SC2) · 5 Jun 2020

Thank you for your interest in the paper. In regards to your first question, are you referring to Table 4 of the paper? If yes; I attached confusion matrices for each method from which you can get an idea of the true positive and false positive rates. In regards to your second question; we left it for future work to do a sensitivity study with more patch sizes.

In regards to Figures 11 and 12; we will update the captions to make it clearer how

to interpret the figures. To help with clarification on how to interpret the figures, let me share with you how we created Figure 11.a and b: We took all the patches that were colocated with CALIOP and selected the patches where CALIOP only detected cirrus/high-altitude clouds over all CALIOP observations that cover the patches. In other words, the patches that were selected exclude patches that had aerosols or just "clear-air" columns or had water clouds present. From the selected patches, we computed how much of each patch that intersects with CALIOP observations were labeled as cirrus/high-altitude cloud by the methods. We then we ranked the fraction results according to cloud optical depth.

The interpretation of Figure 11.a is as follows: For cirrus cloud optical depth of 2, where CALIOP detected only ice clouds in all its observations in the patch, on average the F-CNN method says that ~87% of the patch (that intersects with CALIOP) is covered by ice clouds. We are avoid using the word accuracy in this case, because 1) for a patch the CALIOP observations only covers a line over the patch since CALIOP has a narrow field of view, 2) it is unclear what the CALIOP's absolute accuracy is at detecting ice clouds.

Figure 12 was created in a similar way compared to Figure 11, though with CALIOP aerosol measurements instead of cirrus/high-altitude clouds.

In regards to the description of CNNs; we will add text to clarify what is our research vs. what has already being built in the CNN packages.

Please also note the supplement to this comment:
https://www.atmos-meas-tech-discuss.net/amt-2020-74/amt-2020-74-SC2-supplement.pdf

[Figure]

**Supplement:**

[Figure]

Interpreting row: F-CNNA025 classified 99.92% of the "clear-air" correctly, and 0.08% incorrectly as cumuliform.

---

## Author Comment (AC1) · 8 Jul 2020

We thank the reviewer for the time spent on reading the paper in detail and providing us with helpful feedback. Here are the response to some of the specific comments:

Comment: line 185 I do not see L and K defined. At this point of the paper, the reader gets a bit lost in the maths (at least I did). Please see if you can keep it more discursive and easier to read. I would suggest putting the mathematical details of the algorithms in an appendix.

[Figure]

Response We rewrote parts of sections 2.1.1 and 2.1.2 to make these more accessible to the reader.

Comment: line 223 It might be rather arbitrary to recognize clouds at different altitudes from the True Color images. Could you comment on that. Further down, figure 6 makes this point: how can you be sure that that's a cirrus cloud?

Response: The person who does the labeling through the adapted NASA Worldview website can use the 11um radiances to assess whether there is a high altitude / cirrus cloud. If it is unclear to the person who does the labeling whether a cloud is a cirrus cloud, the cloud top pressure retrievals can also be consulted to help the person make the decision. We added text near line 223 to make this point clearer.

Comment: Section 2.5 is difficult to read. Some details could perhaps be given in an appendix and the text in the paper made more fluid.

Response We attempted to make this section more readable or move it to the appendix.

Comment Section 2.5.3 is a bit too concise. Please expand.

Response This section has been expanded.

Comment: Line 365 Yes, the edges are problematic. Please spend a few words in saying how you will address this problem in future work.

Response We added text to explain how this problem will be addressed in future work.

Comment: Line 428. This is not the focus of this paper, but are there other studies confirming the inconsistency that you see between MODIS and CALIOP data?

Response There are a number of such studies; we added the following citations: "Improving the CALIOP aerosol optical depth using combined MODIS‐CALIOP observations and CALIOP integrated attenuated total color ratio", by Min Oo and Robert Holz, 2011 "Comparison of aerosol optical depth between CALIOP and MODIS-Aqua for CALIOP aerosol subtypes over the ocean", by Man-Hae Kim, Sang-Woo Kim, Soon-

Chang Yoon, and Ali H. Omar, 2013

---

## Author Response (AR1)

**Response to reviewers of paper amt-2020-74**

**Table of content:**

**A. Anonymous Referee #1**

We thank the reviewer for the time spent on reading the paper in detail and providing us with helpful feedback. Here are the response to some of the specific comments:

**1. Comment: line 185**

I do not see L and K defined. At this point of the paper, the reader gets a bit lost in the maths (at least I did). Please see if you can keep it more discursive and easier to read. I would suggest putting the mathematical details of the algorithms in an appendix.

**Response**

The details of the algorithms 1 and 2 have been moved to appendix A. Text were also added that describes L and K.

**2. Comment: line 223**

It might be rather arbitrary to recognize clouds at different altitudes from theTrue Color images. Could you comment on that. Further down, figure 6 makes this point: how can you be sure that's a cirrus cloud?

**Response**

The person who does the labeling through the adapted NASA Worldview website can use the 11um radiances to assess whether there is a high altitude / cirrus cloud. If it is unclear to the person who does the labeling whether a cloud is a cirrus cloud, the cloud top pressure retrievals can also be consulted to help the person make the decision. We added text near line 223 to make this point clearer.

**3. Comment:**

Section 2.5 is difficult to read. Some details could perhaps be given in an appendix and the text in the paper made more fluid.

**Response**

Section 2.5 has been moved to appendix B.

**4. Comment**

Section 2.5.3 is a bit too concise. Please expand.

**Response**

This section has been expanded.

**5. Comment: Line 365**

Yes, the edges are problematic. Please spend a few words in saying how you will address this problem in future work.

**Response**

We added text to explain how this problem will be addressed in future work.

**6. Comment: Line 428.**

This is not the focus of this paper, but are there other studies confirming the inconsistency that you see between MODIS and CALIOP data?

**Response**

A significant source of bias in CALIPSO results from the assumed extinction to backscatter ratio (lidar ratio) used in the aerosol retrieval which can vary significantly for an aerosol type. Fortunately for this research the absolute AOD is not important only that we know that we are looking at a thick aerosol layer. There are a number of such studies; we added the following citations.

1. "Improving the CALIOP aerosol optical depth using combined MODIS‑CALIOP observations and CALIOP integrated attenuated total color ratio", by Min Oo and Robert Holz, 2011
2. "Comparison of aerosol optical depth between CALIOP and MODIS-Aqua for CALIOP aerosol subtypes over the ocean", by Man-Hae Kim, Sang-Woo Kim, Soon-Chang Yoon, and Ali H. Omar, 2013

**B. Anonymous Referee #2**

We thank the reviewer for the time spent on reading the paper in detail and providing us with helpful feedback. Here are the response to some of the specific comments:

**1. Grouped comments**

1) The introduction is somewhat confusing. It's a mixture between an introduction and a method section and thus needs some restructuring, See my specific comments below.

2) P1, L20: Change "Introduction & Problem Solving" to "Introduction".

3) P3, Fig. 1 and P4, Fig. 2: Both figures should be moved to the method section.

4) P3, L64: Skip Sub-header "1.1. Objectives".

5) P5, Section 1.2: Skip section header and move most parts of the text to the

method sections. In the introduction it is sufficient to just write in a few sentences what

6) P5, L1.3: Skip sub header "1.3 Outline".

**Response**

This introduction is, admittedly, a bit unconventional relative to most other scientific papers. However, given the nature of the topic, we think it is important to give some form of explanation and/or foreshadowing to the reader what the proposed methodology is in the introduction. We assume that the reader wants to assess the value of the paper as soon as possible, and therefore determine if the paper is a worthwhile read. We help the reader achieve this goal by including in the introduction a summary of the results via figures 1 and 2, and by giving a short overview of the methodology.

As a compromise, we request to keep figures 1 and 2 in the introduction and move most of the text in section 1.2 to the methods section. Sub-headers "1.1. Objectives", "1.2 Proposed methodology" and "1.3 Outline" will be removed.

**2. Comment: P6, Section 1.4**

This section should be either included in the method section(extra subsection for this is not necessary). Even better would it be to have this Table in an appendix.

**Response**

Since the listed acronyms are used throughout the paper, we would like to make the reader aware of the different commonly used acronyms as soon as possible. This approach is common in many mathematical texts.

**3. Comment: P2, L53**

References? Who has developed the CNN method?

**Response**

References were added.

**4. Comment: P6, L114**

Add a reference for the FE method?

**Response**

After the semicolon the FE algorithm is associated with a CNN.

**5. Comment: P7, L134**

". . .. . ....with which rank the different methods. . .. . ...". Sentence not correct, something is missing here. Did you mean . . .." with which "we" rank the different methods"?

**Response**

This is a grammatical error; the sentence now reads: "...from the test dataset a test (generalization) error is computed with which the different methods are ranked based in their error performance (Friedman et al., 2001)."

**6. Comment: P12, L202**

The listing is obsolete and mentioning the person to do it is a bit weird here. Therefore, I would suggest to rewrite the sentence as follows ". . ....MODIS/VIIRS images that are chosen are divided into overlapping patches. . .. . .."

**Response**

The sentence is changed; thank you for the feedback.

**7. Comment: P12, L205**

"one co-author" obsolete. Simply write "In order to ensure consistent labels for the study, a labeled data set was created ..."

**Response**

We specify "one co-author" to explicitly indicate that (a) all the images were labeled by one person as opposed to an ensemble of labelers and (b) this person is a subject-domain expert. This is salient information given that the creation of this dataset is one of the contributions of the work.

**8. Comment: P12, L207**

Why are quotation marks used for clear air?

**Response**

We are of the opinion that an air column is unlikely to be clear of aerosols. Having the words clear air in quotation marks avoids it being read as an absolute claim.

**9. Comment: P12, Section 2.2.2**

This section sounds like a user manual. Avoid writing the atmospheric scientist/user. Just describe the process itself.

**Response**

The section is being revised to improve clarity, however, we did feel it necessary to be detailed in order for other researchers to be able to replicate our method.

**10. Comment: P13, L225**

". . ..the following guidelines were followed:" Rephrase sentence to avoid repetition of "follow".

**Response**

The sentence is rephrased.

**11. Comment: P22, L416**

Sentence on Fig13 not clear. Please rephrase.

**Response**

The sentence will be rephrased.

**12. Comment: P23, L428**

Are studies existing that compare these instruments and/or document these differences? If yes, please add these references.

**Response**

The following citations were added in which comparisons were made between the MODIS and CALIOP instruments and biases were reported.

1. "Improving the CALIOP aerosol optical depth using combined MODIS‑CALIOP observations and CALIOP integrated attenuated total color ratio", by Min Oo and Robert Holz, 2011
2. "Comparison of aerosol optical depth between CALIOP and MODIS-Aqua for CALIOP aerosol subtypes over the ocean", by Man-Hae Kim, Sang-Woo Kim, Soon-Chang Yoon, and Ali H. Omar, 2013

**13. Comment: P23, L447**

By design of what? The CNN?

**Response**

The human-labeled training dataset only contained optically thick aerosols. Thus, the CNN method learned only to detect thick aerosols. We will rephrase the sentence.

**14. Comments:**

1) P33, Figure 11 caption Some text here belongs rather to the main text since it is describing what conclusions can be drawn from the figures and not what actually is shown.

2) P34, Figure 12 caption: Same here. Shorten the caption text and just write what is shown.

**Response**

The captions are shortened and some of the caption-text was moved to the main text.

**Leveraging spatial textures, through machine learning, to identify aerosol and distinct cloud types from multispectral observations**

Willem J. Marais[1], Robert E. Holz[1], Jeffrey S. Reid[2], and Rebecca M. Willett[3]

[1]Space Science Engineering Center, University of Wisconsin-Madison, Madison, Wisconsin, USA
[2]Marine Meteorology Division, Naval Research Laboratory, Monterey, California, USA
[3]Department of Statistics & Computer Science, University of Chicago, Illinois, USA

**Correspondence:** Willem J. Marais (willem.marais@ssec.wisc.edu)

**Abstract.** Current cloud and aerosol identification methods for multi-spectral radiometers, such as the Moderate Resolution Imaging Spectroradiometer (MODIS) and Visible Infrared Imaging Radiometer Suite (VIIRS), employ multi-channel spectral tests on individual pixels (i.e. field of views). The use of the spatial information in cloud and aerosol algorithms has been primarily statistical parameters such as non-uniformity tests of surrounding pixels with cloud classification provided by the multi-spectral microphysical retrievals such as phase and cloud top height. With these methodologies there is uncertainty in identifying optically thick aerosols, since aerosols and clouds have similar spectral properties in coarse spectral-resolution measurements. Furthermore, identifying clouds regimes (e.g. stratiform, cumuliform) from just spectral measurements is difficult, since low-altitude cloud regimes have similar spectral properties. Recent advances in computer vision using deep neural networks provide a new opportunity to better leverage the coherent spatial information in multi-spectral imagery. Using a combination of machine learning techniques combined with a new methodology to create the necessary training data we demonstrate improvements in the discrimination between cloud and severe aerosols and an expanded capability to classify cloud types. The training labeled dataset was created from an adapted NASA Worldview platform that provides an efficient user interface to assemble a human labeled database of cloud and aerosol types. The Convolutional Neural Network (CNN) labeling accuracy of aerosols and cloud types was quantified using independent Cloud-Aerosol Lidar with Orthogonal Polarization (CALIOP) and MODIS cloud and aerosol products. By harnessing CNNs with a unique labeled dataset, we demonstrate the improvement of the identification of aerosol and distinct cloud types from MODIS and VIIRS images compared to a per-pixel spectral and standard deviation thresholding method. The paper concludes with case studies that compare the CNN methodology results with the MODIS cloud and aerosol products.

**1 Introduction**

**Yellow notes** are responses to reviewer 2's comments
**Red notes** are responses to reviewer 1's comments

[revised manuscript text omitted]